# Superposition Yields Robust Neural Scaling

**Yizhou Liu, Ziming Liu, and Jeff Gore**
Massachusetts Institute of Technology
`{liuyz, zmliu, gore}@mit.edu`

## Abstract

The success of today's large language models (LLMs) depends on the observation that larger models perform better. However, the origin of this neural scaling law, that loss decreases as a power law with model size, remains unclear. We propose that representation superposition, meaning that LLMs represent more features than they have dimensions, can be a key contributor to loss and cause neural scaling. Based on Anthropic's toy model, we use weight decay to control the degree of superposition, allowing us to systematically study how loss scales with model size. When superposition is weak, the loss follows a power law only if data feature frequencies are power-law distributed. In contrast, under strong superposition, the loss generically scales inversely with model dimension across a broad class of frequency distributions, due to geometric overlaps between representation vectors. We confirmed that open-sourced LLMs operate in the strong superposition regime and have loss scaling inversely with model dimension, and that the Chinchilla scaling laws are also consistent with this behavior. Our results identify representation superposition as a central driver of neural scaling laws, providing insights into questions like when neural scaling laws can be improved and when they will break down.[1]

## 1 Introduction

The remarkable success of large language models (LLMs) has been driven by the empirical observation that increasing model size, training data, and compute consistently leads to better performance [1–4]. Across a wide range of tasks — including language understanding [1, 5, 6], math [7–10], and code generation [11, 12] — larger models achieve lower loss, higher accuracy, and greater generalization abilities [2, 13]. This consistent trend, known as neural scaling laws, has been observed across multiple model families and architectures, fueling the development of increasingly large models [2–4]. These scaling laws have not only shaped the current strategies for building better models but have also raised fundamental questions about why such simple and universal patterns emerge in complex learning systems.

The power-law loss with model size plays a central role in both the practical design and the theoretical understanding of large-scale machine learning systems, yet its origin remains inconclusive [3, 14–25]. Various explanations have been proposed, drawing from statistical learning theory and empirical phenomenological models, including improved function or manifold approximation in larger models [14, 15], and enhanced representation or skill learning in larger models [19–22]. In the limit of infinite data, many of these explanations predict a power-law decay of loss with model size, provided the underlying data distribution also follows a power law. The scaling exponents are sensitive to the properties of the data distribution. Moreover, the connection between these mechanistic explanations and the behavior of actual LLMs needs further exploration.

---

[1]Code is available at `https://github.com/liuyz0/SuperpositionScaling`

39th Conference on Neural Information Processing Systems (NeurIPS 2025).

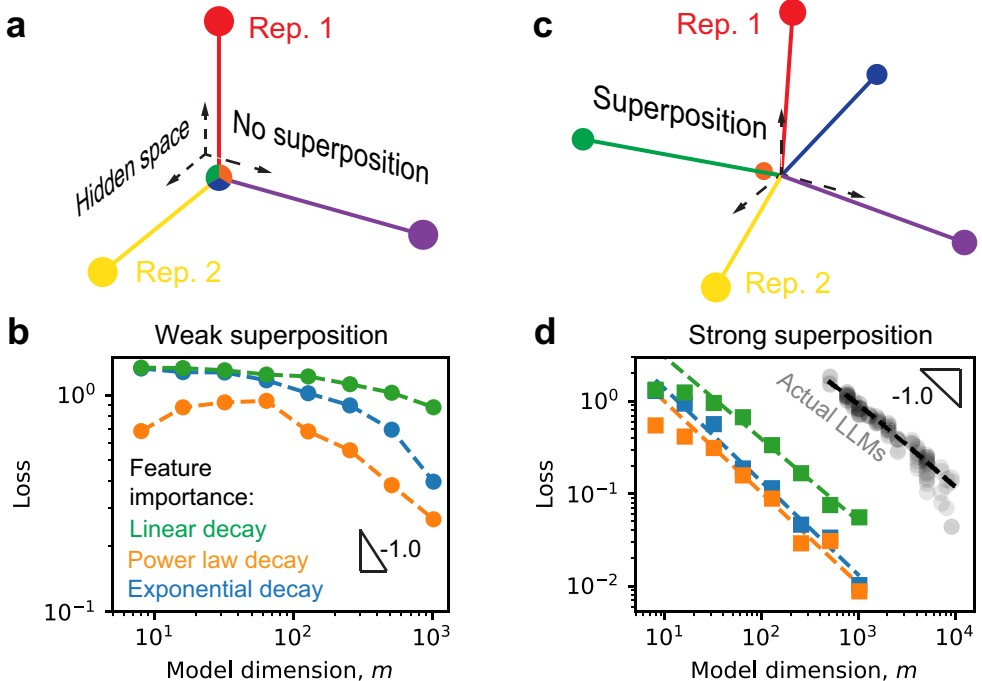

Figure 1: Superposition leads to robust and fast power-law loss decay with model size. (a) Illustration of no superposition where a three-dimensional space can at most represent three features without any interference (overlap). (b) Toy model results in the regime of weak superposition, where we set data dimension $n = 10240$ (number of features). The toy model will be introduced; more details are in Appendix D.1. (c) Illustration of superposition: there are more features than the dimension of the space. (d) The same toy models in the strong superposition regime show lower losses, which are on power laws with model dimension and have exponents close to 1 (color coding same as panel b). The gray points are from actual LLMs, which have a similar power-law exponent near 1.

When considering LLMs specifically, it becomes clear that representation or embedding can be a limiting factor, which is closely related to a phenomenon called **superposition** [26, 27], yet this aspect has not been thoroughly studied. LLMs must learn embedding vectors for tokens, process these representations through transformer layers to predict the next token, and use a final projection (the language model head) to generate the output. Conceptually, fitting functions or manifolds and learning skills or grammars are primarily tasks of the transformer layers, while representation is more directly tied to the embedding matrix and the language model head. To represent more than fifty thousand tokens — or even more abstract concepts — within a hidden space of at most a few thousand dimensions, the quality of representations is inevitably constrained by the model dimension or width, contributing to the final loss. Although models can represent more features than their dimensionality would suggest through a mechanism known as superposition [27], prior works on neural scaling laws seem to fall in the weak superposition regime implicitly [15–20], which may be less relevant to the regime where LLMs operate. This gap leads us to study

> **Question:** How will superposition influence the loss scaling with model dimension (width)?
>
> Varying the degree of superposition and data structure, when is the loss a power law? And if the loss is a power law, what will the exponent be?

We adopt a toy model construction similar to [27] to study how superposition affects neural scaling laws. In the toy model, representations are learned by recovering data, each composed of multiple latent features. These features in data have different frequencies of occurrence, reflecting their relative importance. Weak superposition means that only the most frequent features are perfectly represented, while the others are ignored. As illustrated in Figure 1a, the first three of six features are represented

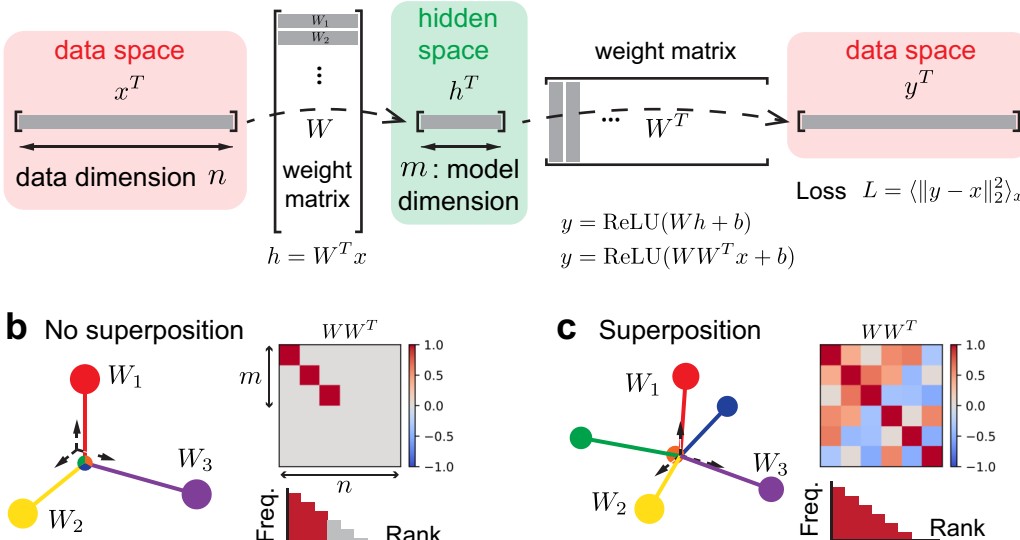

**a** Toy model of representation learning via data recovery

$$y = \text{ReLU}(Wh + b)$$
$$y = \text{ReLU}(WW^T x + b)$$
$$h = W^T x$$

**b** No superposition

**c** Superposition

Figure 2: Toy model of superposition. (a) Architecture and loss of the toy model. (b and c) A row of the matrix $W$, denoted by $W_i$, is the representation of feature $i$. (b) No superposition represented the most frequent features, i.e., the first three ($n = 6$ features in $m = 3$ dimensional space), without interference. In the frequency-rank plot, height means feature $i$'s frequency $p_i$, and color means the $i$th row vector's norm $\|W_i\|_2$. (c) With superposition, features are all represented, while the representations $W_i$ overlap.

in the three-dimensional space without interference, and the remaining three are omitted. We find that in the weak superposition regime, the scaling of loss with model dimension depends sensitively on how feature frequency decays with rank: the loss follows a power law with model size only if the feature frequencies themselves follow a power law, provided that $m$ is sufficiently large (Figure 1b). By contrast, strong superposition allows many more features to be represented, albeit with overlap in the representation (Figure 1c). In this regime, the model displays a robust behavior: loss scales inversely with model dimension across different data frequency distributions (Figure 1d). Remarkably, we find that actual LLMs follow a similar scaling. We summarize our contributions as

**Main results/messages**

- Loss in the weak superposition regime depends on summing frequencies of ignored features, which is a power law if frequencies follow a power law.
- In the strong superposition regime, loss arises from the interference between representations and can have robust "one over width" scaling because of the geometry.
- LLMs exhibit strong superposition and agree quantitatively with our toy model predictions.

The rest of the paper will elaborate on the takeaways. In Section 2, we introduce the toy model, describe the data sampling procedure, and explain how we control the degree of superposition. Section 3 presents the detailed results. In Section 4, we compare our findings to related works. Finally, Section 5 summarizes our conclusions and discusses limitations and future directions.

## 2 Methods

To understand the relationship between superposition and data structure, we need a toy model to represent data features simple enough yet not simpler — two key principles need to be reflected, (i) there are more features to represent than the dimension of the model, and (ii) features occur in data with different frequencies. Later, we will discuss how the loss due to representation studied here may affect the overall final loss in LLMs.

We adopt the toy model of superposition from Anthropic [27] (an autoencoder) with minor modifications (Figure 2a). Input $x \in \mathbb{R}^n$ is a vector with data dimension $n$ being the number of atomic (or irreducible) features. Each element $x_i$ in $x$ is interpreted as the activation of this sample at feature $i$, which follows

$$x_i = u_i v_i, \; u_i \sim \text{Bernoulli}(p_i) \; \& \; v_i \sim U(0, 2). \tag{1}$$

Here, $u_i$ sampled from a Bernoulli distribution controls whether the feature $i$ is activated, and $v_i$ sampled from a uniform distribution controls the activation strength once feature $i$ is activated. All samples are i.i.d. The frequency of feature $i$ to appear in the data is $p_i$. Without loss of generality, we make the indices of features the same as their frequency or importance rank. The data structure is then about how $p_i$ decreases with rank $i$. The expected number of activations in one input will be referred to as activation density: $E = \sum_{i=1}^{n} p_i$. The model learns hidden representations by recovering the data, which cannot be done perfectly because the model dimension $m$ is much smaller than the number of possible features in the data $n$. The trainable parameters are a weight matrix $W \in \mathbb{R}^{n \times m}$ and a bias vector $b \in \mathbb{R}^n$. The weight matrix embeds data $x$ into a hidden space with dimension $m$, $h = W^T x$, with $m \ll n$. In practice, we fix $n$ as a large number and change the model dimension $m$. We use $W$ to read out the embedding, where $y = \text{ReLU}(Wh + b)$. The loss is defined as the difference between the recovered $y$ and the original $x$, $L = \langle \|y - x\|_2^2 \rangle_x$, where $\langle \cdot \rangle_x$ means average over $x$ distribution.

We can now formally introduce superposition. Note that $W_i$ is the representation of feature $i$ in the hidden space, where we use $W_i$ to denote the $i$th row of the $W$ matrix. We emphasize the following

> **Key concepts**
>
> • Feature frequency: $p_i$ is the probability that feature $i$ is activated (non-zero) in a sample, which is assumed to decrease with $i$.
> • Sparsity: We say features are sparse when $E/n$ is small.
> • The feature $i$ is represented (in the hidden space) when $W_i$ is non-zero.

No superposition ideally means the first $m$ rows of $W$ form an orthogonal basis (i.e., the first $m$ most important features represented perfectly) and the rest of the rows are zero (i.e., the rest of the features ignored or lost), as illustrated in Figure 2b. Superposition means that there are more than $m$ rows in $W$ with non-zero norms (Figure 2c).

We next summarize important facts of this toy model [27]:

> **Preliminaries**
>
> Superposition cannot lead to lower losses in a linear model (without ReLU function) due to the large amount of interference. ReLU and negative biases can cancel off interference, realizing error correction. With this non-linearity, superposition can be preferred when feature frequencies are more even (better not to ignore features) and features are sparse in data (error correction is easier).

We can see that in Figure 1, where features are sparse in data, the losses in the strong superposition regime are indeed much smaller than those in the weak superposition regime across several feature frequency distributions.

If one regime is more preferred, we want to approach it more quickly in training. If it is not preferred, we also want to study the scaling behaviors scientifically in that regime. To this end, we introduce a decoupled weight decay (or growth) term in training to tune the degree of superposition:

$$W_{i,t+1} = \begin{cases} W_{i,t} - \eta_t \gamma W_{i,t}, \; \gamma \geq 0, \\ W_{i,t} - \eta_t \gamma W_{i,t}(1/\|W_{i,t}\|_2 - 1), \; \gamma < 0, \end{cases} \tag{2}$$

where $\eta_t$ is the learning rate and $W_{i,t}$ is the $i$th row of the weight matrix at step $t$ (vector operations are element-wise). For weight decay $\gamma < 0$, the update corresponds to gradient descent on $(\|W_{i,t}\|_2 - 1)^2$, encouraging unit-norm rows. We implement this weight decay in AdamW [28] optimizer with a warm-up and cosine decay learning rate schedule (details in Appendix B). At each training step, we sample new data.

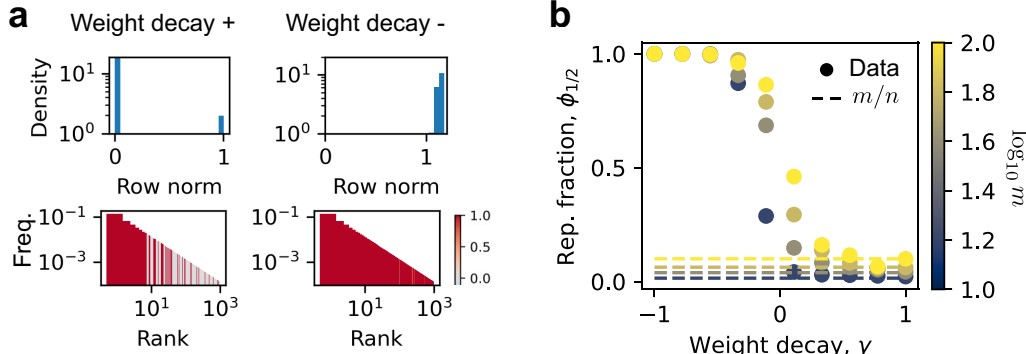

Figure 3: Weight decay can tune the degree of superposition. (a) Positive weight decay ($\gamma = 1$ in the figure) has $\|W_i\|_2$ near 0 or 1, with frequent features more likely to be represented (color means $\|W_i\|_2$ in frequency-rank plots). Negative weight decay ($\gamma = -1$) has $\|W_i\|_2$ around 1. We show results when $\alpha = 1$, $m = 100$, yet the claim is generally true. (b) For all models, small weight decays lead to strong superposition, and large weight decays lead to no superposition ($\phi_{1/2} \approx m/n$). More data in Appendix D.3.

We find that the weight decay can robustly control superposition. We first see that important features tend to be represented (associated $\|W_i\|_2 > 0$), and norms of $W_i$ become bimodal, clustering near 0 or 1 (Figure 3a). This allows us to define the fraction of represented features as

$$\phi_{1/2} = |\{i : \|W_i\|_2 > 1/2\}|/n, \tag{3}$$

namely, the fraction of rows with norm larger than $1/2$.[2] We found that weight decay can tune superposition for all models we trained, with small weight decay $\gamma$ giving strong superposition, i.e., $\phi_{1/2} \approx 1 \gg m/n$, and large weight decay corresponding to weak superposition, i.e., $\phi_{1/2} \sim m/n$ (Figure 3b). The ability of weight decay to tune superposition is robust to feature frequency distributions (Appendix D.3). We can then systematically study scaling behaviors in different regimes.

The toy model differs from LLMs in architecture, data, and loss. Since we focus on representations rather than next-token prediction, we omit transformer layers. Conceptually, LLMs map a document to a token, with inputs and outputs in different spaces, while the toy model operates within a single shared space. Despite this, the toy model captures key aspects of language structure through engineered sparsity and feature importance, making its data structure aligned with that of LLMs at a high level. While LLMs use cross-entropy loss and the toy model uses squared error, we can show that this does not affect the scaling behaviors (Appendix A.2). Thus, the toy model is a suitable abstraction for studying representation-limited scaling.

## 3 Results

For a systematic scan, we set $p_i \propto 1/i^\alpha$ in this section and can vary the **data exponent** $\alpha$ to change how skewed $p_i$ is.[3] The **activation density** $E$ is set as 1, whose value can be shown to not affect the scaling (Appendix D.4). We fix **data dimension** $n = 1000$, vary **model dimension** $m$ from 10 to 100, and sweep **weight decay** $\gamma$ from $-1$ to 1. We fit final test losses as a power law, $L \propto 1/m^{\alpha_m}$, and call $\alpha_m$ the **model exponent**. More details on hyperparameters are in Appendix B.2.

### 3.1 Weak superposition

We seek to understand when loss follows a power law with model dimension, and what determines the exponent when it does in the weak superposition regime. Consider an idealized case where the top $\phi_{1/2}n$ most frequent features are perfectly represented (no overlap), where $\phi_{1/2}$ is the fraction of

---

[2]In theory, we should use 0 as the threshold. The choice, $1/2$, may minimize misclassifications since norms are near 0 or 1. Our result is robust to this threshold since norms are very concentrated.

[3]The word or phrase frequency in natural language follows Zipf's law, which is a power law ($\alpha = 1$).

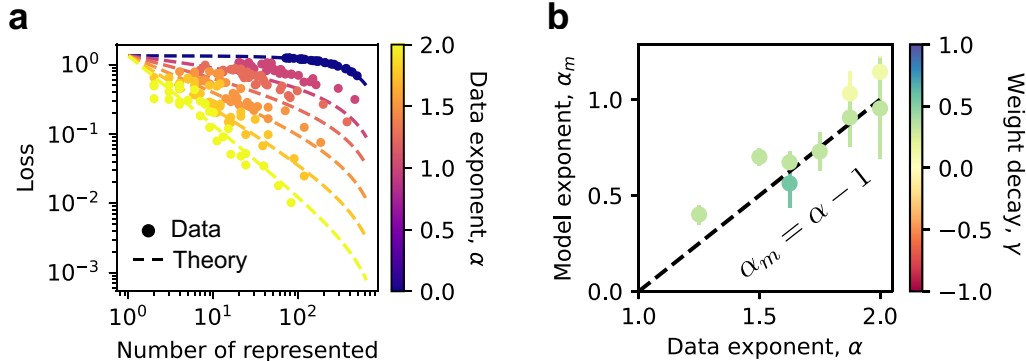

Figure 4: Loss at weak superposition can be well described by the frequency sum of ignored features. (a) Observation and theory at weak superposition (i.e., Equation (4) as a function of number of represented features, $\phi_{1/2}n$) agree when weight decay $\gamma$ is positive. (b) For those closest to the ideal no superposition case, we expect $\alpha_m = \alpha - 1$, which is close to measured values. Error bars are standard errors. Details in Appendix D.5.

represented features. The optimal biases are $b_i = 0$ for $i \leq \phi_{1/2}n$ and $b_i = \langle x_i \rangle$ for $i > \phi_{1/2}n$ [27]. The loss can then be written as:

$$L = \sum_{i > \phi_{1/2}n} \langle (x_i - \langle x_i \rangle)^2 \rangle = \sum_{i > \phi_{1/2}n} (\langle v^2 \rangle p_i - \langle v \rangle^2 p_i^2) \approx \langle v^2 \rangle \sum_{i > \phi_{1/2}n} p_i. \tag{4}$$

The last approximation is right when $p_i \ll 1$ for $i > \phi_{1/2}n$ and $p_i^2$ terms are negligible. We use the definition of $x_i = u_i v_i$, where $v \sim U(0, 2)$, giving $\langle v^2 \rangle = 4/3$. We can use the integral $\int_{\phi_{1/2}n}^{n} p_i \mathrm{d}i$ to estimate the summation, yielding an expression that depends on the number of represented, $\phi_{1/2}n$, and the data exponent, $\alpha$ (Appendix D.5). We find that, in the weak superposition regime, the actual losses closely match this prediction (Figure 4a). Focusing on cases closest to the ideal no-superposition scenario, where $\phi_{1/2}n = m$ and the first $m$ features are represented, we observe that such cases occur when $\alpha > 1$ and yield a model exponent $\alpha_m \approx \alpha - 1$ (Figure 4b). This matches the theoretical expectation that $\int_m^n p_i \mathrm{d}i \propto m^{-\alpha+1}$ when $n \gg m$ and $\alpha > 1$. Thus, in the weak superposition regime, loss scaling is well described by the contribution of unlearned features, that is, the total frequency of features not represented by the model.

We can now answer our Question in the weak superposition regime.

> **Result 1: "Power law in, power law out" in the weak superposition regime**
>
> The loss is governed by a sum of frequencies of less frequent and not represented features. Ideally, there are model dimension $m$ most important features being represented. If feature frequencies follow a power law, $p_i \propto 1/i^\alpha$ with $\alpha > 1$, the loss or the summation starting at $m$ will be a power law with $m$ with exponent $\alpha - 1$.

This finding of the specific toy model agrees with previous works with very different settings [15–20], where some power-law skill importance or spectrum is assumed.

## 3.2 Strong superposition

We next turn to the strong superposition regime. Consider the case where only feature $j$ is activated. The output $y_j$ has activation $\sim W_i \cdot W_j$, leading to a loss that scales as squared overlaps $(W_i \cdot W_j)^2$ due to the definition of loss. The loss arises from the non-zero overlaps between representation vectors. We cannot solve the weight matrix $W$ in this regime. The section goes back and forth between theoretical ansatz and experimental observations to understand the high-level behaviors.

We start by considering relatively even feature frequencies, where trained $W_i$ are expected to be isotropic. One simplest theoretical ansatz of isotropic vectors is i.i.d. vectors uniformly on the unit sphere. In $\mathbb{R}^m$, the squared overlap of two such random vectors follows $\text{Beta}(\frac{1}{2}, \frac{m-1}{2})$ distribution,

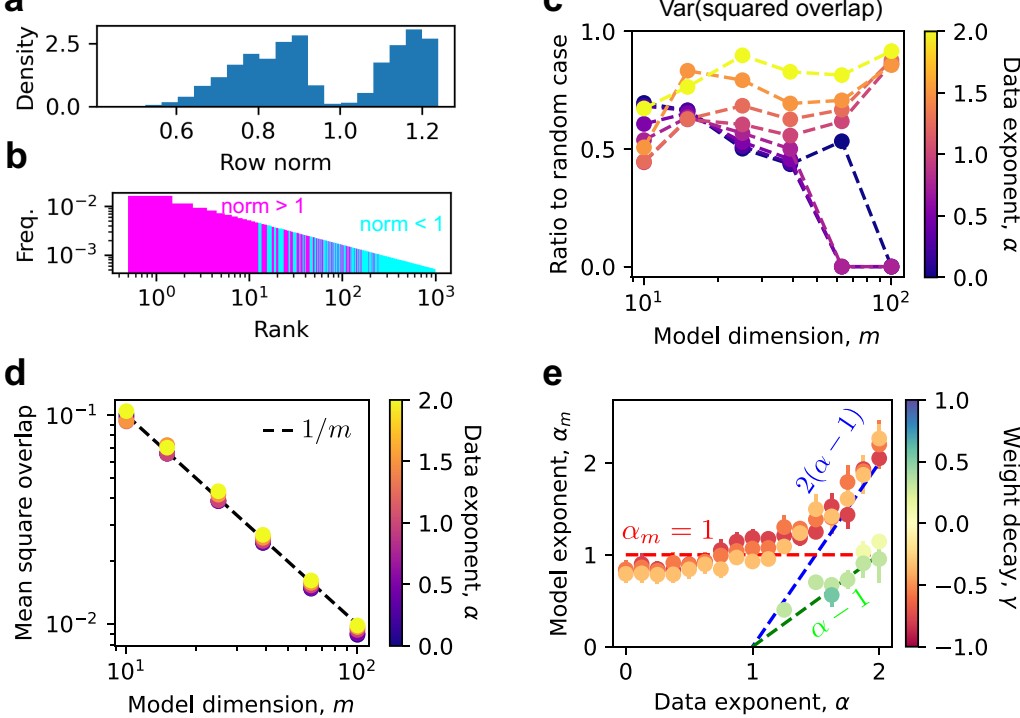

Figure 5: Loss scaling at strong superposition is explained via geometry. (a) The row norm distribution is bimodal around $1$. (b) The more frequent the features are, the more likely their norms are greater than $1$. (c) Variance of squared overlaps for features with $\|W_i\|_2 > 1$ is smaller than that of random unit vectors, i.e., $\frac{2(m-1)}{m^2(m+2)}$. Overlaps are calculated using directions $W_i/\|W_i\|_2$. We show the measured variances ($\gamma = -0.55$) divided by the above theory value for random vectors. (d) The features with $\|W_i\|_2 > 1$ have $1/m$ mean squared overlaps, where we plotted all the data when $\gamma < 0$. (e) At strong superposition ($\gamma < 0$), $\alpha_m = 1$ if feature frequencies are flat ($\alpha$ small) due to isotropic vector geometry. But $\alpha_m \approx 2(\alpha - 1)$ if the feature frequencies are skewed ($\alpha$ large). Error bars are standard errors. More details in Appendix D.6.

and therefore has mean $1/m$ and variance $\frac{2(m-1)}{m^2(m+2)} \sim 2/m^2$. The squared overlaps for isotropic random vectors typically obey $1/m$ scaling.

The actual trained $W_i$ have structures whose norms are bimodal near $1$ (Figure 5a), and more important features tend to have larger vector norm (Figure 5b). We want to understand how such a structure will change the scaling of overlaps. It turns out that for better error correction (using bias to cancel interference), the model needs to minimize the maximum overlap rather than the sum of squared overlaps. Consider $\nu$ unit vectors $w_i \in \mathbb{R}^m$ with $\nu \geq m$. It can be shown that [29]

$$\max_{i \neq j} |w_i \cdot w_j| \geq \sqrt{\frac{\nu - m}{m(\nu - 1)}} \equiv \kappa. \tag{5}$$

The lower bound, $\kappa \approx \sqrt{1/m}$ when $\nu \gg m$. The bound is met when the vectors form an equal angle tight frame (ETF) [30–32], which has no variance in absolute overlaps and appears in contexts such as quantum measurements [33–36] and neural collapse [37, 38]. ETFs in real spaces can only exist if $\nu \leq \frac{m(m+1)}{2}$ [30–32]. We find that the $W_i$ with $\|W_i\| > 1$ associated with important features tend to be ETF-like (Figure 5, c and d): the variance of squared overlaps is smaller than that of random vectors and can be near $0$ for even feature frequencies (small $\alpha$); the mean of squared overlaps collapse on $1/m \approx \kappa^2$. Being ETF or ETF-like can help error correction and reduce loss values, but would not change the typical scaling with $m$ if the number of vectors is much larger than $m$. Similar to ETFs, whose number of vectors is bounded, the number of vectors $W_i$ with $\|W_i\| > 1$ is around $m^2/2$ (Appendix D.6), and the less important features tend not to be represented (norm lower than $1$).

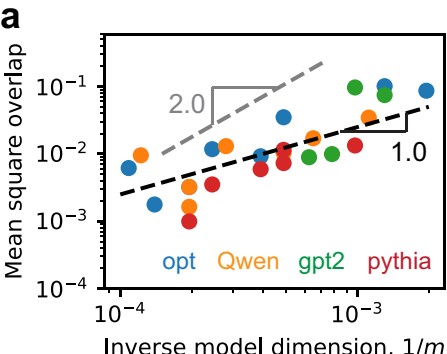 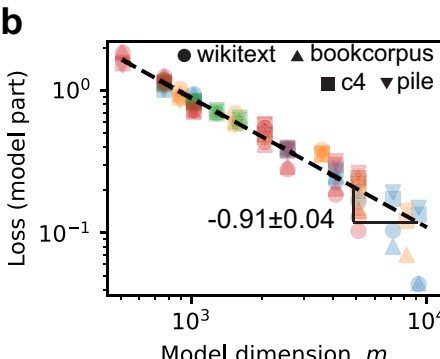

Figure 6: Superposition may explain the neural scaling law observed in actual LLMs. We evaluate four open-sourced model classes, Opt [39], GPT2 [40], Qwen [41], and Pythia [42], which have model sizes from around 100M to 70B (evaluation details in Appendix C). (a) We found the mean square overlaps of $W_i/\|W_i\|_2$ roughly follow $1/m$ scaling, where $W$ is the language model head. (b) The model class is reflected by color as panel a, while we use shapes for evaluation datasets [43–46]. The loss related to model size is fitted as a power law, yielding empirical $\alpha_m = 0.91 \pm 0.04$ close to 1. More analysis in Appendix D.7.

Vectors of these less important features cannot be explained with a simple theoretical ansatz. Yet, combining the lessons from random vectors and ETF, we expect the squared overlaps to scale as $1/m$ robustly for isotropic vectors (confirmed in Appendix D.6). Considering all the overlaps when feature frequencies are even, we then predict the loss to scale as $1/m$, which is true (Figure 5e).

When the feature frequencies are skewed (large $\alpha$), we find that the model exponent $\alpha_m$ increases with $\alpha$ and becomes greater than 1. Vectors being non-isotropic, i.e., important features having much smaller overlaps, may lead to larger $\alpha_m$. To illustrate this idea, we conjecture an extreme situation where the $m^2/2$ most important features can be ETF-like and contribute negligible loss compared to the less important ones. In the worst case (Appendix A.1), the less important features lead to a loss proportional to $\sum_{i=m^2/2}^{n} p_i \sim m^{-2(\alpha-1)}$, i.e., $\alpha_m = 2(\alpha-1)$, which is close to observations (Figure 5e). The real configuration of $W_i$ is more complicated than this simple conjecture, requiring advanced future studies on when $\alpha_m$ loses robustness and how it depends on feature frequencies sensitively then. To conclude the section, we have

> **Result 2: Geometric origin of $1/m$ loss scaling ($\alpha_m = 1$) at strong superposition**
>
> For even feature frequencies, vectors $W_i$ tend to be isotropic in space with squared overlaps scaling like $1/m$ when $n \gg m$, leading to the robust $1/m$ power-law loss. For skewed feature frequencies, representation vectors are heterogeneous in space, making loss sensitive to feature frequencies, where it might need power-law frequencies to have power-law losses.

### 3.3 LLMs

Finally, we explore how our findings might be relevant to real LLMs [39–42]. As a naive mapping, we treat tokens as atomic features, with data dimension $n$ equal to the vocabulary size. The model dimension $m$ for LLMs is known. We analyze the language model head, denoted by the weight matrix $W$. Through the norm and interference distributions of the rows of $W$, we claim LLMs are in superposition (Appendix D.7). If we measure token frequency, it follows a power law with exponent $\alpha$ close to 1 (Appendix D.7). We conclude, based on the knowledge from toy models, that LLMs operate in a superposition regime, and expect loss to be related to squared overlaps $\sim 1/m$. We next calculated the mean squared overlaps of normalized rows $W_i/\|W_i\|_2$, and found they roughly obey $1/m$ scaling (Figure 6a). We argue that cross-entropy loss, given that the overlaps are small in absolute value, can be expanded and approximately scales as the mean square overlaps (Appendix A.2). We therefore expect the loss of representation-limited LLMs to have $1/m$ scaling. LLM losses are close to a linear function of $1/m$ (Appendix D.7). Yet when $m \to \infty$, the extrapolation of losses does not

hit 0. The non-zero intersection can be due to intrinsic uncertainty in language. Increasing model sizes decreases "wrong" interferences but cannot eliminate uncertainty in the data. So, as in previous papers where loss is decomposed into model size part, dataset size part, and a constant [3], we fit our loss values by the following,

$$L = C_m/m^{\alpha_m} + L_{\setminus m}, \tag{6}$$

where the model size part $C_m/m^{\alpha_m}$ is universal (model size is a function of $m$), and $L_{\setminus m}$ contains loss irrelevant to model size, depending on the evaluation dataset and model class. The fitting yields $\alpha_m = 0.91 \pm 0.04$ (Figure 6b). We inferred from the Chinchilla models [3] that due to model size $N \propto m^{2.52 \pm 0.03}$ (Appendix D.7), $\alpha_m = (2.52 \pm 0.03) \times \alpha_N = 0.88 \pm 0.06$, where $\alpha_N = 0.35 \pm 0.02$ [47] is the power-law exponent of loss with model size. The exponents $\alpha_m$ from LLMs are close to 1. We highlight the finding as

---

**Result 3: Superposition is an important mechanism behind LLM neural scaling laws**

LLMs operate in the strong superposition regime. The squared overlaps of token representations scale as $1/m$, token frequencies are flat ($\alpha = 1$), and the model size relevant loss scales closely to $1/m$, agreeing with the toy model prediction.

---

## 4 Related works

Neural scaling laws were first characterized empirically [2], demonstrating that for LLMs, the cross-entropy loss improves predictably as a power-law with increased model size (parameters), dataset size, or compute, over multiple orders of magnitude. This finding is built on earlier observations (e.g. [48]) that deep learning performance scales in a smooth power-law fashion with data and model growth. Many works showed the surprisingly universal nature of such scaling behaviors across architectures and tasks [2–4], directing further development of LLMs.

Several heuristic toy models have been proposed to explain neural scaling laws. One common view is that models aim to fit data manifolds or functions, and the scaling exponents depend strongly on the structure of the data [14, 15]. Another group of models assumes the network learns discrete features or skills [19, 20], whose importance follows a power-law distribution, giving results the same as ours in the weak superposition regime. One toy model predicts that loss scales inversely with model width [25], arguing that parameters independently perform the same task with noise, and the scaling follows from the central limit theorem. However, this model applies in the overparameterized cases and may be less relevant to LLMs.

More formal approaches rely on similar heuristics. The scaling behavior depends on how the dataset size and model size approach infinity. When the dataset is fixed and model size grows to infinity, the system is variance-limited, and loss scales as $1/m$ by central limit theorem arguments [15]. When the dataset size grows to infinity first, the loss scaling enters the resolution-limited regime. In linear models or kernel methods, this leads to $\alpha_m = \alpha' - 1$ [15–18], seemingly consistent with our weak superposition regime. Here, $\alpha'$ is the exponent of the power-law decay of kernel eigenvalues, which can be seen as abstract feature importance. Considering neural tangent kernels, $\alpha'$ depends on both data and the model configuration. Our work may be framed as mechanistically showing that $\alpha' = \alpha$ ($\alpha$ is the intrinsic data exponent) when models have no superposition, and $\alpha'$ is something else when models have strong superposition, which is new. The resolution-limited regime has also been described as fitting the data manifold [15].

Our toy model is based on Anthropic's model of superposition [27] (an autoencoder), with modifications to the data sampling. The original study explored how data structure influences superposition but did not explicitly control it. Related models have appeared in compressed sensing [49–53] and neural information processing [54, 55], yet with distinct contexts and objectives. Besides representation, people also studied calculation in superposition [56, 57].

## 5 Discussion

Our work is built on observations of the toy model and analysis without rigorously solving the toy model. We are thus limited to explaining deeper behaviors in the toy model. Our analysis of LLMs

suggests they are in the strong superposition regime, but the underlying reasons were not studied in detail. We believe one reason is that features are sparse in language, as the number of tokens required to predict one token is much less than the total number of tokens. The softmax function may also be important since it is strong at error correction, giving superposition an advantage.

Neural scaling laws also include scaling laws with dataset size and with training steps, which we did not study. At each step, a fixed number of new data points are used for optimization. So, we expect the scaling with the total data amount and that with training steps will be the same, similar to the results at weak superposition [20]. However, in the strong superposition regime, data or training step scaling is related to angle distribution and how angles between representations evolve, which cannot be easily explained without rigorous solving.

We focused on representation loss, yet LLMs should also have losses due to parsing or processing in the transformer layers. We imagine that the loss associated with model size can be written as

$$C_m/m^{\alpha_m} = f_m(m) + f_\ell(\ell), \tag{7}$$

where $\ell$ is the depth of the LLM, $f_m$ and $f_\ell$ are two functions capturing the loss due to representation and parsing, respectively. A future direction is to study the parsing-limited scaling (i.e, $f_\ell(\ell)$ function) independently. It is also plausible that the observed scaling of inference time [58] is connected to this parsing-limited regime. We here write the equality because $\ell$ depends on $m$ in LLMs [39–42]. Given model size $N$, $m$ and $\ell$ are constrained (roughly, $N \propto m^2\ell$). There is an optimal $m$-$\ell$ relationship such that the loss $f_m(m) + f_\ell(\ell)$ can be minimized given $N$ [59]. At this optimal $m$-$\ell$ relationship, $f_m(m)$ and $f_\ell(\ell)$ should be balanced. Therefore, we expect $f_\ell(\ell)$ to be similar to $f_m(m)$. And if $f_m(m) \sim 1/m$ due to superposition and $f_\ell(\ell)$ is similar, we can measure an empirical $\alpha_m \approx 1$ from data, which is true. Or, if the width-limited loss is much larger, we can also observe that the total loss due to model size has $\alpha_m \approx 1$. We conclude that superposition in any case is an important mechanism underlying neural scaling laws.

Beyond explaining existing phenomena, our results may offer guidance for future LLM development and training strategies:

> Assuming our explanation of width scaling is correct, we ask **can we change the loss scaling with width to be faster than power laws, or to have larger exponents**? The answer is **no** for natural languages but may be **yes** for domain tasks with super skewed feature frequencies. Another question is **when the scaling law will stop**? Based on our naive connection between features and tokens, the answer is that when the model dimension reaches the vocabulary size, the loss limited by width will deviate from a power law and vanish. However, the vocabulary size may set a lower bound for the true number of independent things in language, then the power law with width may continue for a longer time.

Recognizing that superposition benefits LLMs, encouraging superposition could enable smaller models to match the performance of larger ones (with less superposition) and make training more efficient. Architectures such as nGPT [60], which constrain hidden states and weight matrix rows to the unit sphere (promoting superposition), demonstrate improved performance. Optimizers that stabilize training without weight decay have also shown promising results [61], potentially due to enhanced superposition. Yet, these improvements may be related to altering coefficients in the neural scaling laws rather than the exponents. We also acknowledge that encouraging superposition may cause difficulties for the mechanistic interpretation of models and AI safety [27, 62].

As a side note, with the same pre-training loss, LLMs with different degrees of superposition may exhibit differences in emergent abilities such as reasoning or trainability via reinforcement learning [63], requiring future studies.

In conclusion, we studied when loss can be a power law and what the exponent should be with different data properties and degrees of superposition. We found that geometric interference at strong superposition may explain the LLM neural scaling laws observed [3]. Our results contribute to a deeper understanding of modern artificial intelligence systems, which also open various directions for future research. We hope our insights will support the continued development and training of more capable and efficient LLMs.

## Acknowledgments and Disclosure of Funding

We are grateful for feedback and suggestions from Yasaman Bahri, Cengiz Pehlevan, Surya Ganguli, Blake Bordelon, Daniel Kunin, and the anonymous reviewers. The authors acknowledge the MIT Office of Research Computing and Data for providing high performance computing resources that have contributed to the research results reported within this paper. J. G. thanks the Sloan Foundation for funding. The authors declare no competing interests.

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

# A Theoretical analysis

## A.1 Toy model loss

We provide a simple analysis for toy model loss scaling. The expected loss in the weak superposition regime is well explained by Equation (4). We do not need to repeat it here.

In the strong superposition regime, we consider an even special data sampling, where data $x$ is sampled such that each data point has and only has one activated feature. The frequency for feature $i$ to be activated is still $p_i$. After determining which feature is activated, say $i$, we still sample $x_i$ as $v_i$ from $U(0,2)$. This sampling is different from the experiments. Yet, since we learned that activation density does not affect scaling exponent (Figure 14), we expected this analysis to predict at least the scaling exponent. Under the assumptions, we have

$$L = \sum_{i=1}^{n} p_i \Big\langle \sum_{j\neq i} \mathrm{ReLU}^2(W_j \cdot W_i v_i + b_j) + (\mathrm{ReLU}(W_i \cdot W_i v_i + b_i) - v_i)^2 \Big\rangle_{v_i}. \tag{8}$$

We are unable to solve for the optimal $W$ and $b$ such that this loss $L$ is minimized. Yet, it is easy to see that we want $\|W_i\|_2^2$ to be close to 1, $W_j \cdot W_i$ to be as small as possible, and $b_j$ to be small negative values of the same order of magnitude as $W_i \cdot W_j$, such that the interference terms $\mathrm{ReLU}(W_j \cdot W_i v_i + b_j)$ may vanish and the recovered feature value $\mathrm{ReLU}(W_i \cdot W_i v_i + b_i)$ can be close to the real one $v_i$.

For convenience, based on the observation that the vector norms are bimodal around 1 in the strong superposition regime, we define strongly represented features as those that have $\|W_i\|_2 > 1$, which are more frequent and ETF-like, and weakly represented ones for those with $\|W_i\|_2 < 1$. We can quantify the fraction of strongly represented as

$$\phi_1 = |\{i : \|W_i\|_2 > 1\}|/n, \tag{9}$$

which significantly exceeds $m/n$ and is around the ETF expectation $m^2/2n$ (Figure 16).

We now review our conjectured extreme configuration, consisting of strongly represented and weakly represented features. The first $\phi_1 n$ most important features are considered to be strongly represented, whose absolute overlap with any other representation scales as $\sqrt{1/m}$. The rest of the features are weakly represented and squeezed into a small angle such that they have small overlaps with the strongly represented, while they can have large overlaps with each other. With such a configuration, the first $\phi_1 n$ terms in the summation of Equation (8) will scale as $1/m$ since each term $\langle \cdots \rangle_{v_i}$ scales as $1/m$. The rest of the terms in Equation (8), in the worst scenario that the terms $\langle \cdots \rangle_{v_i}$ do not decrease obviously with $m$, will be proportional to $\sum_{i=\phi_1 n}^{n} p_i$. If the strongly represented features dominate, we can have $1/m$ scaling for the loss. On the contrary, when the weakly represented dominate, we expect the loss to have scaling like $\sum_{i=\phi_1 n}^{n} p_i$. More specifically, if $p_i \sim 1/i^\alpha$ ($\alpha > 1$) and we use $m^2/2$ to approximate $\phi_i n$, the loss scales as $1/m^{2(\alpha-1)}$.

## A.2 Cross-entropy loss

We provide the reason why cross-entropy loss also scales as squared overlaps. We consider the last hidden state after going through the normalization layer is $W_i/\|W_i\|_2$, such that the output should be the $i$th token. By constructing such an example, we ignore possible loss due to parsing but focus on the loss just due to representation. The loss from this data point is

$$L = -\ln \frac{e^{\|W_i\|_2}}{\sum_j e^{W_i \cdot W_j/\|W_i\|_2}} = \ln \Big[1 + \sum_{j\neq i} e^{W_i \cdot W_j/\|W_i\|_2 - \|W_i\|_2}\Big]. \tag{10}$$

We assume that $W_i \cdot W_j/\|W_i\|_2$ is much smaller than 1 since we know the overlap scale as $1/m$. We then approximate the loss via Taylor expansion

$$L = \ln \Big[1 + (n-1)e^{-\|W_i\|_2} + \sum_{j\neq i}[W_i \cdot W_j/\|W_i\|_2 + (W_i \cdot W_j/\|W_i\|_2)^2/2]e^{-\|W_i\|_2}\Big]. \tag{11}$$

In the first thought, the summation $\sum_{j\neq i} W_i \cdot W_j$ should be zero since there are positive and negative overlaps distributed evenly if the vectors span the whole space. But in language, one sentence can

have different continuations, connecting different tokens. For example, both putting "cats" or "dogs" after "I like" are legit. The existence of data "I like" then will tend to squeeze different tokens closer to each other. The summation $W_i \cdot W_j$ should be a small positive constant $\epsilon_{D,i}$ related to the correlation in data. The reason we keep the second-order term is clear now as they are the lowest order terms related to model sizes. We keep expanding the $\ln$ function and have

$$L = (n-1)e^{-\|W_i\|_2} + \frac{\epsilon_{D,i}e^{-\|W_i\|_2}}{\|W_i\|_2} + \frac{1}{2}\sum_{j\neq i}\left(\frac{W_i \cdot W_j}{\|W_i\|_2}\right)^2 e^{-\|W_i\|_2}. \tag{12}$$

The part related to the model size is mainly

$$L_m = \frac{1}{2}\sum_{j\neq i}\left(\frac{W_i \cdot W_j}{\|W_i\|_2}\right)^2 e^{-\|W_i\|_2}. \tag{13}$$

In this construction, one can see that once $\|W_i\|_2$ is sufficiently large, the loss can be arbitrarily low, which does not happen in reality. The reason is still related to the intrinsic uncertainty in language data. If one sentence can have different continuations, we need in the hidden space, a region that can lead to large probabilities over different tokens. However, when the norm is too large, one will find that the hidden space is sharply separated — each hidden state yields high probability only on one token. We then expect the norm $\|W_i\|_2$ to be as large as possible such that $ne^{-\|W_i\|_2}$ is small while $\|W_i\|_2$ is upper bounded by intrinsic data uncertainty. Therefore, $\|W_i\|_2$ should not depend on model size much (verified in Appendix D.7). The loss related to model size $L_m$ then scales as $1/m$ since the cosine similarity scales as $\sqrt{1/m}$ and $L_m$ is related to the squared cosine similarity in the lowest order approximation.

## B   Toy model training

In this Appendix, we explain how we trained the toy models and obtained raw data. There are two classes of toy models trained. The first one is a large toy model with data dimension $n = 10240$, which is reported in Figure 1 and Figure 9 to show scaling behavior across around two orders of magnitude. The other toy model class is small toy models fixing $n = 1000$, such that we can scan more hyperparameters. Figures 3, 4, 5, 14, and 9 use small toy models.

### B.1   Large toy models

We implemented a neural network experiment to study the scaling of feature representation and recovery. The toy model is defined as a two-layer neural network with ReLU activation (see Figure 2).

The hyperparameters are given as follows.

- Data dimension $n$: 10240
- Model dimension $m$: Varied exponentially from $2^3$ to $2^{10}$
- Batch size: 2048 (tested up to $8192$, which does not affect final loss)
- Total training steps: 20000 (tested up to $80000$, which does not affect final loss)
- Learning rate: Initially set to 0.02, scaled according to hidden dimension
- Weight decay: $-1.0$ for strong superposition, and $0.1$ for weak superposition
- Device: Training performed using one V100 GPU, with floating-point precision (FP32)

Data points $x$ were synthetically generated at each training step according to Equation (1) to simulate feature occurrence frequencies. We considered three distributions with activation density $E = 1$:

- Exponential: $p_i \propto e^{-i/400}$
- Power-law: $p_i \propto i^{-1.2}$
- Linear: $p_i \propto n - i$

We employed the AdamW optimizer with distinct learning rates and weight decay settings for the weight matrix $W$ and bias vector $b$. Specifically, for weight matrix $W$, learning rate was scaled as lr $\times (8/m)^{0.25}$ with specified weight decay. And for bias vector $b$, a learning rate of $2.0/m$ was used with no weight decay. A cosine decay learning rate schedule with a warm-up phase (5% of total steps) was implemented. At each training step, input data batches were dynamically generated based on the selected probability distribution. The final test loss is calculated across newly sampled data with a size being 100 times the batch size.

The model and optimizer were compiled and executed on a CUDA-enabled GPU for efficient training. After training, weight matrices $W$ and training losses were stored and analyzed.

Final outputs, including weight matrices and training loss histories, were saved in PyTorch format for subsequent analysis and visualization.

This setup provided a structured exploration of feature representation scaling under varying dimensions and distributions, crucial for understanding superposition and scaling laws in neural networks.

The code can be found in `exp-17.py`.

## B.2  Small toy models

We conducted numerical simulations using a neural network model designed for feature recovery. The objective was to analyze the model's behavior across various conditions involving feature frequency skewness (controlled by data exponent $\alpha$), model dimensions, and weight decay parameters.

In the small toy models reported in Figures 3, 4, 5, 14, and 9, we set the hyperparameters as

- Feature dimension $n$: Fixed at 1000.
- Hidden dimension $m$: Varied logarithmically between 10 and 100 , across 6 distinct sizes, i.e., $m = 10, 15, 25, 39, 63, 100$.
- Batch size: 2048.
- Training steps: 20000 steps for each condition.
- Learning rate: Initialized at $1 \times 10^{-2}$, dynamically adjusted using cosine decay scheduling with a warm-up phase of 2000 steps.
- Weight decay: Explored systematically from -1.0 to 1.0, in increments of 0.22 approximately (10 discrete values).
- Data exponent $\alpha$: Ranged linearly from 0 to 2, with 17 discrete steps.

In Figure 14, we fix data exponent $\alpha = 1$ while scan 9 activation densities linearly from 1 to the maximal value $\sum_{i=1}^{n} 1/i$. All other settings are the same.

Synthetic data was generated for each batch based on a power-law probability distribution, defined as:

$$p_i \propto \frac{1}{i^\alpha} \quad \text{where} \quad i \in \{1, 2, \ldots, n\}$$

with the condition $\sum_i p_i = E$. Each element of the batch data $x$ was randomly activated based on this probability, then scaled by a uniform random value between 0 and 2.

At each training step, input batches were regenerated, and the learning rate was updated following the cosine decay schedule described above.

The training performance was evaluated using Mean Squared Error (MSE) loss computed between the network output and input batch data at every step. Final weights were saved for further analysis. The final test loss is calculated across newly sampled data with a size being 100 times the batch size.

The simulations were performed in parallel using 96 CPU cores, where each core executed one distinct parameter combination defined by the weight decay and data exponent values. Or, in Figure 14, the parameter combination is defined by weight decay and activation density values.

Loss histories and trained weight matrices were saved separately for post-experiment analysis. Files were systematically indexed to indicate the corresponding experimental parameters. This detailed setup facilitated a comprehensive investigation of model behavior under diverse training and data distribution conditions.

The code can be found in `exp-10.py`, `exp-10-3.py`, and `exp-15.py`.

## C   LLM evaluation

### C.1   Overlap analysis

We analyzed the row overlaps of the language model head weight matrices among various large language models (LLMs) to investigate the geometric properties of their hidden spaces.

We selected models from the following families, varying widely in parameter count:

- OPT (from OPT-125m to OPT-66b)
- Qwen2.5 (from 0.5B to 72B)
- GPT-2 (GPT2, GPT2-Medium, GPT2-Large, GPT2-XL)
- Pythia (from 70m to 12B)

Weights were downloaded directly from Hugging Face model repositories. For each model, the weight matrix or language modeling head was normalized by its row norms:

$$W_i \leftarrow \frac{W_i}{\|W_i\|_2 + \epsilon}, \quad \epsilon = 10^{-9},$$

where $\epsilon$ is for numerical stability.

We computed the pairwise absolute cosine overlaps between all normalized vectors using batch-wise computations for efficiency. The overlap between embedding vectors $W_i$ and $W_j$ is given by:

$$\text{overlap}(W_i, W_j) = \left| \frac{W_i \cdot W_j}{\|W_i\|_2 \|W_j\|_2} \right|.$$

To handle large embedding matrices efficiently, overlaps were computed in batches (size of 8192 vectors).

We calculated two key statistics for the overlaps within each model:

- Mean Overlap: The average of absolute overlaps for all unique vector pairs:

$$\text{mean\_overlap} = \frac{\sum_{i<j} \text{overlap}(W_i, W_j)}{n(n-1)/2}$$

- Overlap Variance: Calculated as:

$$\text{variance\_overlap} = \frac{\sum_{i<j} (\text{overlap}(W_i, W_j) - \text{mean\_overlap})^2}{n(n-1)/2}$$

From these values, we can calculate mean square overlaps as $\text{mean\_overlap}^2 + \text{variance\_overlap}$.

The calculations were accelerated using GPU resources (CUDA-enabled) to efficiently handle computations involving extremely large matrices.

Results including mean overlaps, variances, and matrix dimensions were recorded for comparative analysis across model sizes and architectures.

The code is in `overlap-0.py`.

### C.2   Evaluation loss

This experiment aims to evaluate multiple large language models (LLMs) efficiently using model parallelism and dataset streaming techniques. The models were assessed on standard text datasets to measure their predictive performance systematically.

Models were selected from Hugging Face and evaluated using a model-parallel setup:

- OPT series

- Qwen2.5 series
- GPT-2 series
- Pythia series

We used the following publicly available datasets for evaluation:

- Wikitext-103: Standard English language modeling dataset.
- Pile-10k: A subset of The Pile, designed for diverse textual data.
- C4: Colossal Clean Crawled Corpus, containing large-scale web text.
- BookCorpus: Large-scale collection of books used for unsupervised learning.

Datasets were streamed directly, efficiently sampling 10000 text segments with a maximum sequence length of 2048 tokens ($\sim 2 \times 10^7$ tokens).

Texts from datasets were tokenized using the respective model-specific tokenizers. Tokenization involved truncation and manual padding to uniform batch lengths. Specifically, padding tokens were assigned an ID of 0, and label padding utilized a special token (-100) to ensure they did not contribute to loss computations.

Each model was loaded using Hugging Face's AutoModelForCausalLM with model parallelism enabled, allowing the evaluation of large models that exceed single-GPU memory limits. Evaluations were conducted in batches, employing a DataLoader with a custom collate function for optimized memory use.

The model's predictive performance was assessed by computing loss values internally shifted by the Hugging Face library, suitable for causal language modeling.

Model parallelism was implemented to efficiently distribute computations across multiple GPUs, leveraging CUDA-enabled hardware.

For each model and each dataset, we run one evaluation and save the evaluation losses.

Random seeds and deterministic sampling ensured reproducible dataset selections, though explicit seed settings were noted as commented options within the implementation.

Evaluation results, including loss metrics and potentially intermediate model states, were systematically stored for detailed post-analysis.

The code is in `cali-1.py`.

## C.3 Token frequency

The purpose of this analysis is to compare token frequencies generated by different tokenizers across several widely-used textual datasets. Understanding these frequencies helps in assessing the representational capacity and efficiency of tokenizers used by various large language models.

We considered the same four datasets mentioned for LLM evaluation. And we use four different tokenizers from the four model classes we evaluated.

Each tokenizer processed textual data from the specified datasets, streaming data directly to efficiently handle large-scale inputs. A target of 1,000,000 tokens per tokenizer-dataset pair was set to ensure sufficient statistical representativeness.

For each dataset-tokenizer combination:

1. Text samples were streamed directly from the datasets.
2. Text was tokenized without adding special tokens (e.g., EOS).
3. Token frequencies were counted and accumulated until the target token count (1 million tokens) was reached.
4. Token frequencies were saved as JSON files for subsequent detailed analyses.

Token frequency data was systematically stored for each tokenizer and dataset combination, enabling comparative analyses of token distributions. The data files provide foundational insights into tokenizer efficiency and coverage across diverse textual domains.

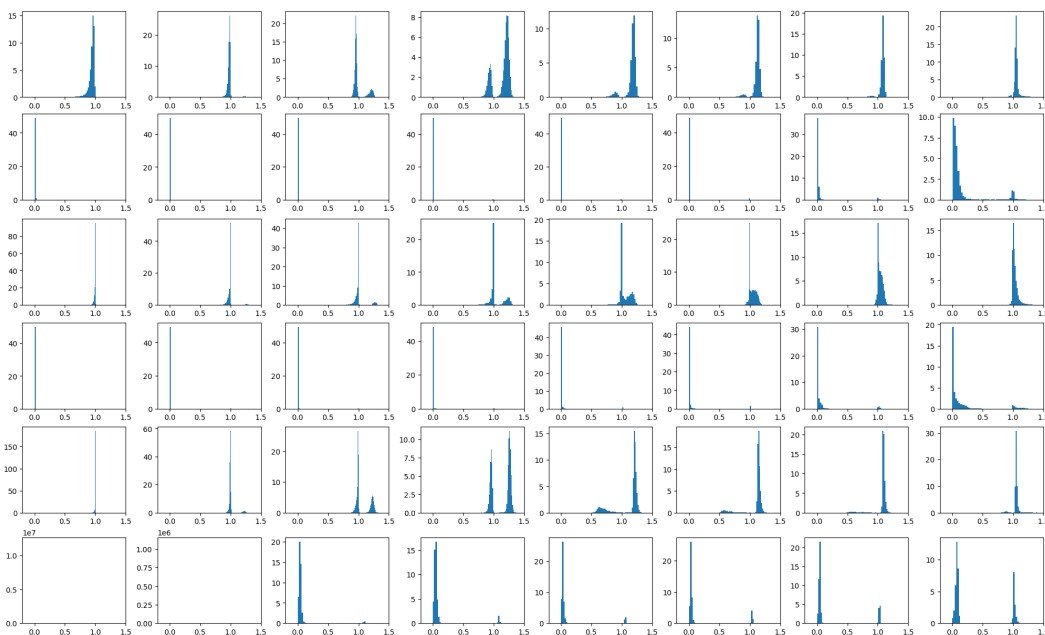

Figure 7: Row norm distributions for large toy models. There are 6 rows of panels. Rows 1 and 2 correspond to the power-law feature frequency. Rows 3 and 4 correspond to the exponential feature frequency. Rows 5 and 6 correspond to the exponential feature frequency. In the two rows that correspond to the same feature frequency, the upper row is at strong superposition and the lower one is at weak superposition. The 8 columns from left to right correspond to different model dimensions $m$ from small to large.

The code is in `token-freq-0.py`.

## D    Figure details and supplementary results

Here, we show how to process the raw data obtained from toy models or LLMs to generate results seen in the main text. Some supplementary analysis is also conducted to support the main text arguments.

### D.1    Figure 1

The toy models reported in Figure 1 are large toy models with data dimension $n = 10240$ explained in Appendix B.1. After obtaining the final losses, we directly plot them with respect to the model dimension $m$. Error bars are calculated as the standard deviation of losses over 100 batches. The error bars are smaller than the dots (Figure 1, b and d).

When we are fitting the loss in log-log as a line, we choose the linear part to fit. If the loss versus model dimension curve is obviously not a line, we fit the whole curve as a line and output the $R^2$ value as a measure of how non-linear it is. Specifically, we fit the last five points for the power-law decay feature case in the weak superposition regime (yellow data in Figure 1b). Other cases in the weak superposition regime are fitted to a line with all data. In the strong superposition regime, when feature frequency decreases as a power law or as a linear function, we fit the data as a line starting from the third point (yellow and green in Figure 1d). And for exponential decay feature frequencies, we fit all the data with a line. In the strong superposition regime, the measure model exponent $\alpha_m$ are close to 1: $1.01 \pm 0.05$ (exponential decay), $1.0 \pm 0.1$ (power-law decay), and $0.89 \pm 0.05$ (linear decay).

The LLM data are copied from Figure 6b, with slope $-0.91 \pm 0.04$ being close to 1 as well. We will explain details about Figure 6 later.

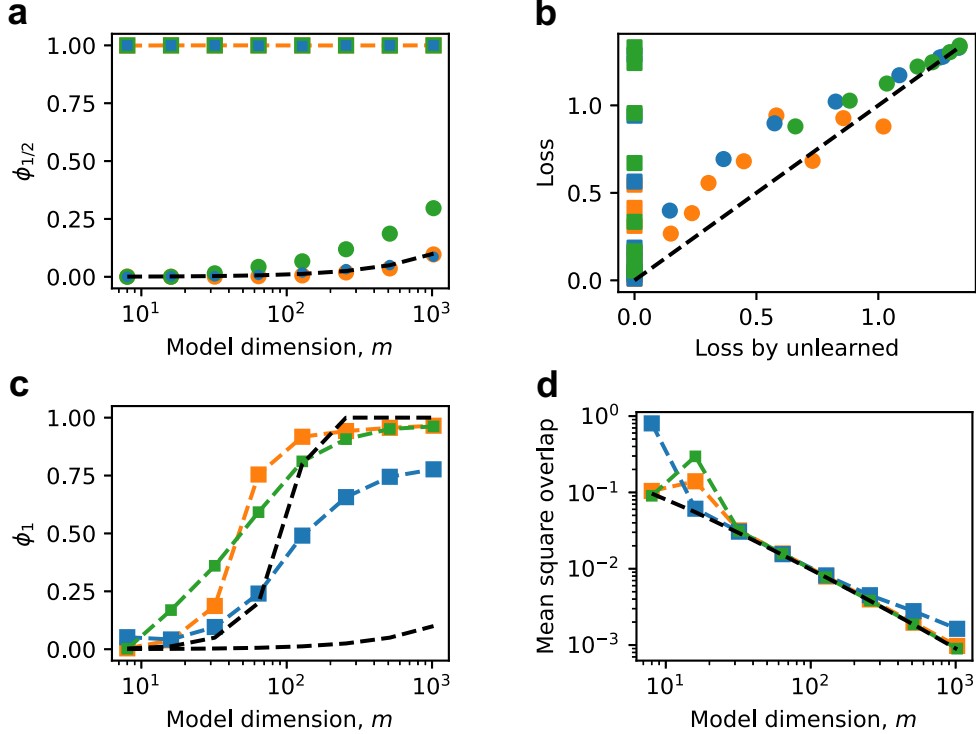

Figure 8: Large toy models agree with theoretical expectations. We use blue, yellow, and green for exponential, power-law, and linear feature frequencies, respectively. Dots correspond to the weak superposition regime, and squares to the strong superposition regime. (a) The fraction of represented features is 1 for strong superposition, and is close to $m/n$ (black dashed line). (b) The expected number of activated but unlearned features well describes the loss at weak superposition. The dashed line is where the actual loss is the same as the predicted one. (c) In the strong superposition regime, the number of strongly represented features is much larger than $m$ but bounded by some value around $m^2/2$. The slowly growing dashed line is $m/n$, and the fast growing dashed line is $\min\{1, m^2/2n\}$. (d) The mean squared overlap of strongly represented features is close to $\kappa^2$, which is close to $1/m$, given that the number of strongly represented features is much larger than $m$.

We also output the weight matrix $W$ for these large toy models (Figure 7). They follow the same pattern that in the weak superposition regime, row norms are bimodal and are either close to 0 or 1, making 0.5 a good separation point for measuring how many features are represented. And in the strong superposition regime, the row norms are distributed near 1, and 1 is a good separation point for the two peaks, i.e., the peak greater than 1 refers to strongly represented features which are more important, and the peak smaller than 1 corresponds to the weakly represented.

We can analyze the large toy model in the same way as what has been done in Figure 4 and 5. The fraction of represented features $\phi_{1/2}$ is calculated, which is 1 in the strong superposition regime, while it is close to $m/n$ in the weak superposition regime (Figure 8a).

With the measured $\phi_{1/2}$, we can estimate the loss due to unlearned features, $\langle v^2 \rangle \sum_{i=\phi_{1/2}n}^{n} p_i$. This theoretical value agrees well with the actual loss in the weak superposition regime (Figure 8b).

In the strong superposition regime, the fraction of strongly represented features is calculated, agreeing with the expectation that the number of strongly represented features is much larger than $m$ but bounded by some value around $m^2/2$ (Figure 8c).

At the end, we see that the mean square overlap of the strongly represented is close to the characterized value $\kappa^2$ (Figure 8d), which scales as $1/m$ since the number of the strongly represented is much larger than $m$.

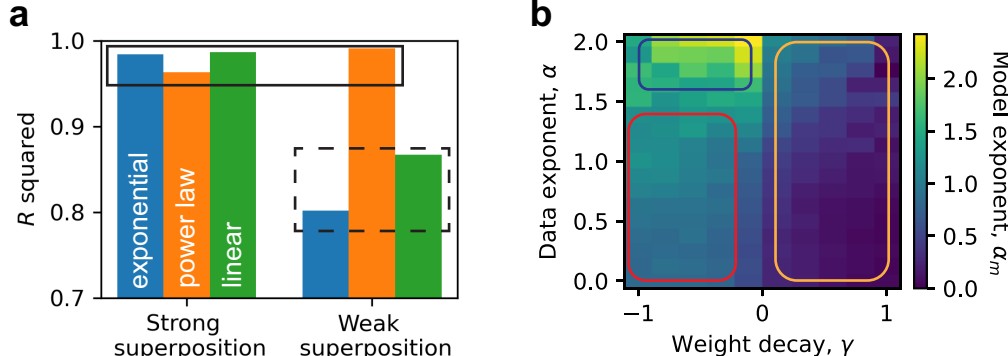

Figure 9: Rich scaling phenomena arise when we change the degree of superposition and data structures. (a) In the same experiments ($n = 10240$) in Figure 1, where $p_i \propto \exp(-i/400)$ (blue), $p_i \propto 1/i^{1.2}$ (yellow), and $p_i \propto n - i$ (green), we use $\gamma = -1$ to reach strong superposition and $\gamma = 0.1$ for weak superposition. R-squared value of the fitting is used to measure how likely the fitted part is a power law (Figure 10). R-squared values closer to 1 mean that the data are more similar to a power law. We found that at strong superposition, power laws are robust across different underlying feature distributions. Yet, at weak superposition, only power-law feature frequencies can lead to power-law losses. (b) When changing superposition by weight decay and varying feature frequency decay by $\alpha$ given $p_i \propto 1/i^{\alpha}$, we found roughly three distinct behaviors. For (b), $n = 1000$ and $m = 10, 15, 25, 39, 63, 100$.

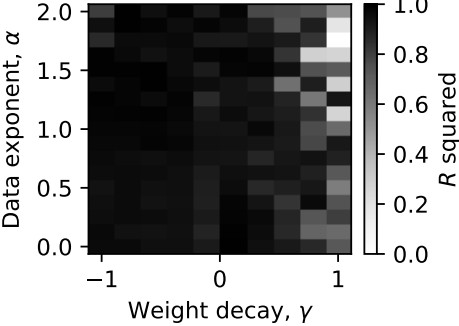

Figure 10: R squared values for fitting loss as a power law with model dimension. Data are from the small toy models with data dimension $n = 1000$.

In Figure 1, we set weight decay $\gamma = -1$ to have strong superposition and $\gamma = 0.1$ to have weak superposition. We compute $R$ squared values from linear fits in log-log plots to quantify scaling behavior, assessing how closely the loss follows a power law to model dimension. We can see that at strong superposition, the losses are close to power laws, regardless of the underlying feature frequencies, yet the loss is a power law at weak superposition if the feature frequency $p_i$ is a power law with rank $i$ (Figure 9a). For a systematic scan, we next set $p_i \propto 1/i^{\alpha}$ and can vary the data exponent $\alpha$ to change how fast $p_i$ decays consistently. Assuming a power-law form for the final test loss, $L \propto 1/m^{\alpha_m}$, we extract the model exponent $\alpha_m$ from the empirical fit. We fit the loss with a power law in all cases. The fitted $\alpha_m$ reveals how fast losses decay, even in the regime where the loss should not be a power law. Roughly, three distinct patterns emerge: (1) under weak superposition (positive $\gamma$, yellow box in Figure 9b), $\alpha_m$ is small, indicating slow loss decay; (2) under strong superposition and a wide range of small data exponents (red box), $\alpha_m$ remains robustly near 1; (3) for strong superposition with large data exponents (blue box), $\alpha_m$ increases with $\alpha$. By interpreting these three patterns, we aim to understand when loss follows a power law with model dimension, and what determines the exponent when it does.

Figure 9a reports the $R^2$ values from the fitting, where the raw data comes from training large toy models (Appendix B.1). When we are fitting the loss in log-log as a line, we choose the linear part to

fit. If the loss versus model dimension curve is obviously not a line, we fit the whole curve as a line and output the $R^2$ value as a measure of how non-linear it is. Specifically, we fit the last five points for the power-law decay feature case in the weak superposition regime (yellow data in Figure 1b). Other cases in the weak superposition regime are fitted to a line with all data. In the strong superposition regime, when feature frequency decreases as a power law or as a linear function, we fit the data as a line starting from the third point (yellow and green in Figure 1d).

And from the raw data of small toy models (Appendix B.2), we can fit the model exponent $\alpha_m$ directly and plot it as a function of $\gamma$ and $\alpha$ as in Figure 9b.

The fitting in Figure 9b does not care whether the loss versus model dimension curve is a power law or not. We provide the $R$ squared values for the fitting here (Figure 10). The closer $R$ squared values are to 1, the better the data can be thought to be a power law (a line in log-log plot). In the strong superposition regime, the $R$ squared values suggest the data are close to be power-law. While in the weak superposition regime, data may not be power-law, especially when $\gamma$ is too large. When $\alpha$ is smaller than 1, it is not a power law in theory. The $R$ squared values are not too small since the loss decay is very slow, and a line in log-log plot is still a good approximation. When $\alpha > 1$ and $\gamma \approx 1$, the number of represented features can be smaller than $m$ or even non-increasing. Too large weight decay still makes the configuration of the representation be in no superposition. However, it destroys some feature representations that can exist, making the configuration far from the ideal case where $m$ features are represented. So, we may not see power laws when weight decay is too strong.

## D.2   Figure 2

Figure 2 introduced the toy model and the concept of superposition without real data. The $W$ matrix we used to show superposition in Figure 2c is obtained by optimizing the square of off-diagonal terms of the normalized $W$, i.e., each row is normalized to have norm 1 first.

## D.3   Figure 3

In Figure 3, we reported results from the trained small toy models with data dimension $n = 1000$, whose detailed hyperparameters are in Appendix B.2.

We showed results at data exponent $\alpha = 1$ in Figure 3. The results are obtained at $m = 100$, $\gamma = -1$ for panel a, and at different $m$ and $\gamma$ for panel b. We showed that the more frequent features tend to have larger norms or to be better represented. And the norm distribution is very bimodal. We here show that it is true that the norm is around 1 or 0 for various $\alpha$ and model sizes $m$ and degrees of superposition (Figures 11 and 12). The fraction of represented, $\phi_{1/2}$, can be calculated directly.

Here, we provide the heat map of $\phi_{1/2}$ at different $m$ as a function of $\alpha$ and $\gamma$ (Figure 13). The pattern is robust, suggesting weight decay is a good tool to change the degree of superposition regardless of data properties and model sizes.

## D.4   Sparsity does not affect scaling behaviors in our tests

We studied the effect of the number of expected activated features or activation density $E$, which was set to 1. By fixing data exponent $\alpha = 1$, which will be shown to be relevant to natural language, we can scan different superposition degrees and activation densities. Since $p_i \leq 1$ is required, which is equivalent to $p_1 \leq 1$, we have $E \leq \sum_{i=1}^{n} 1/i^\alpha$, setting the upper bound for our scanning. We found that loss is approximately proportional to activation density $E$ (Figure 14a). This fact suggests that the power law exponent should not change, which we confirmed (Figure 14b). Under a controlled superposition degree, activation density linearly increases loss and thus does not affect the scaling exponents in our experiments.

Once obtaining the small toy models scanning activation density and keeping $\alpha = 1$, we can plot the loss as a function of activation density $E$ in Figure 14. The linear fitting is also straightforward. We chose one $\gamma$ to show in the main text. Here, we present the whole picture that, with any weight decay tested, the model exponent is robust to the change of activation density (Figure 15).

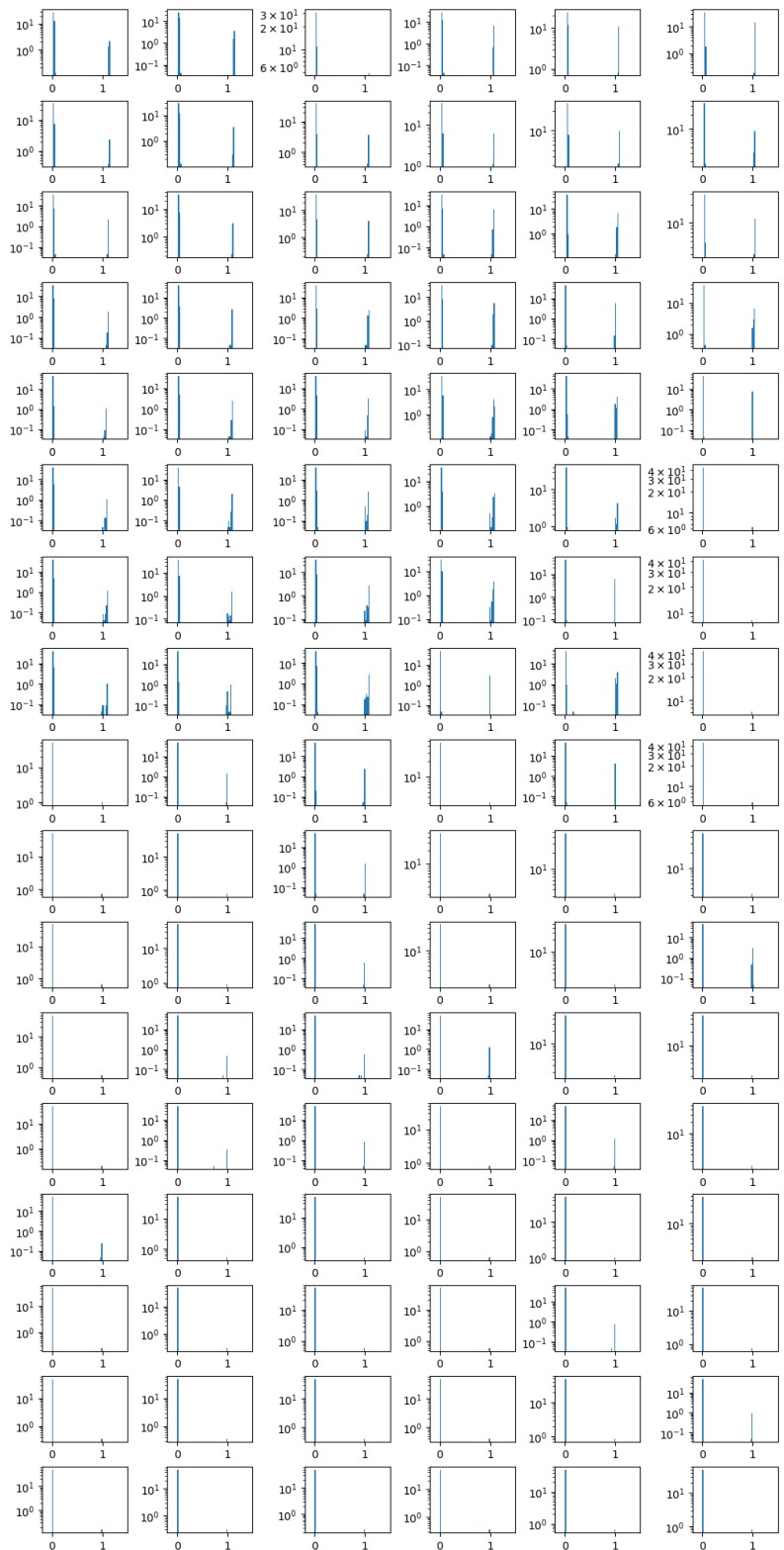

Figure 11: Row norm distribution at weak superposition ($\gamma = 0.55$) shows that the rows either are close to zero or have norm close to 1, making $0.5$ a good separation. The 17 rows of panels from top to down correspond to 17 $\alpha$ from 0 to 2. And the 6 columns from left to right correspond to $m = 10, 15, 25, 39, 63, 100$.

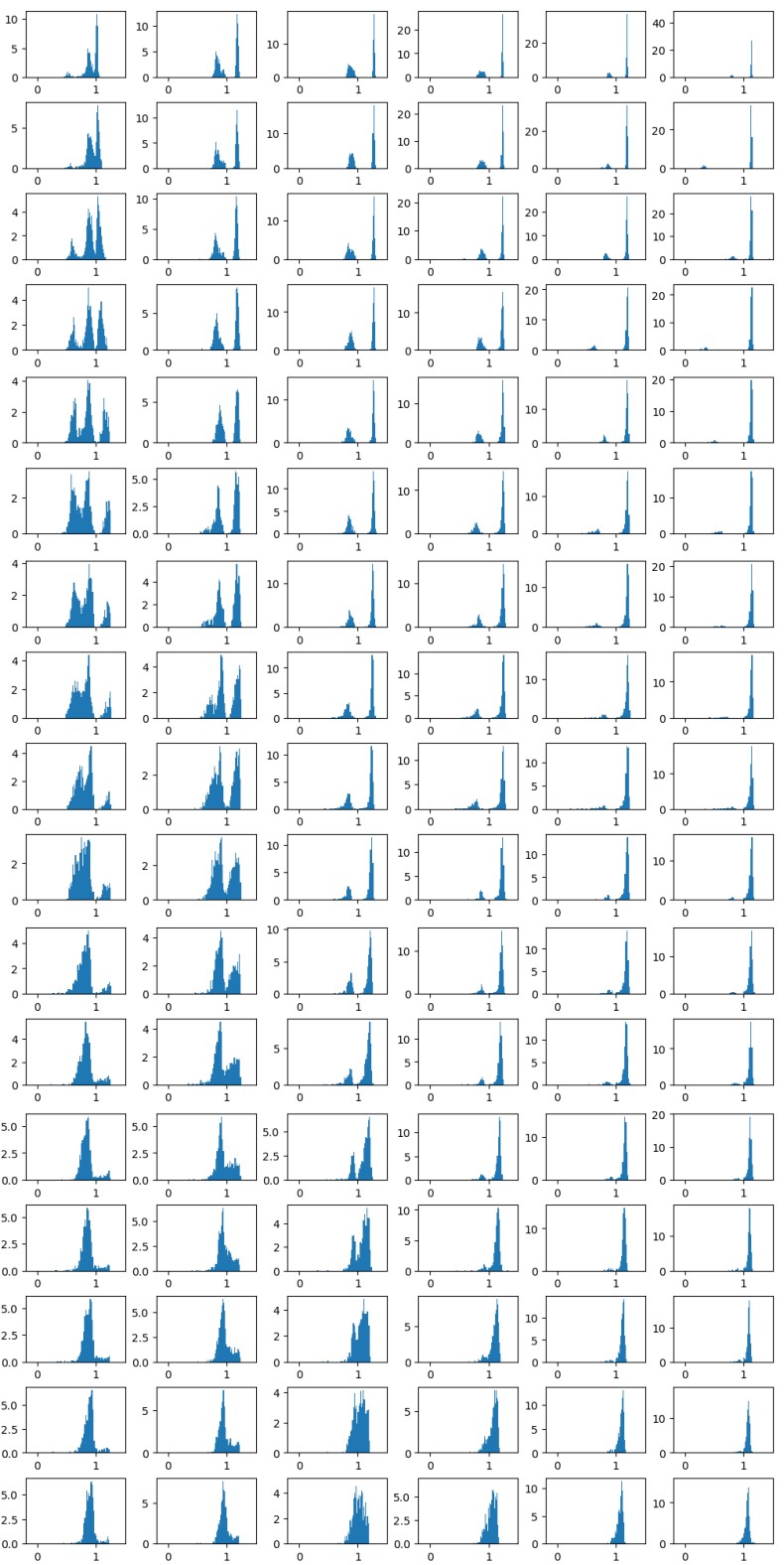

Figure 12: Row norm distribution at strong superposition ($\gamma = -0.55$) shows that the rows have norm close to 1. And density at 1 is very low, making 1 a good separation point for two groups of row norms. The 17 rows of panels from top to down correspond to 17 $\alpha$ from 0 to 2. And the 6 columns from left to right correspond to $m = 10, 15, 25, 39, 63, 100$.



Figure 13: Fraction of represented features as a function of $\gamma$ (x-axis) and $\alpha$ (y-axis). The 6 columns from left to right correspond to $m = 10, 15, 25, 39, 63, 100$. The colorbar is $\phi_{1/2}$ where purple means 1 and white means 0.

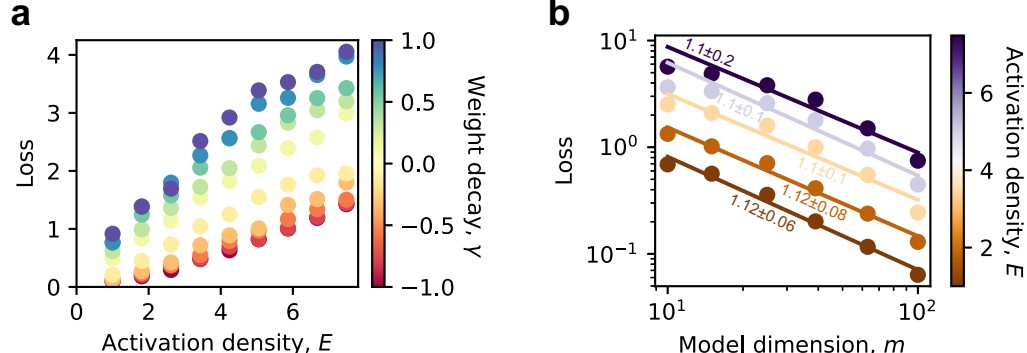

Figure 14: Activation density does not affect scaling exponents in our tests. (a) Loss is roughly proportional to activation density given the degree of superposition ($m = 63$, $n = 1000$). (b) So, $E$ will only affect the coefficient but not the exponent when considering the power law with model dimension. We plot the evidence $\alpha_m \approx 1$ at strong superposition.

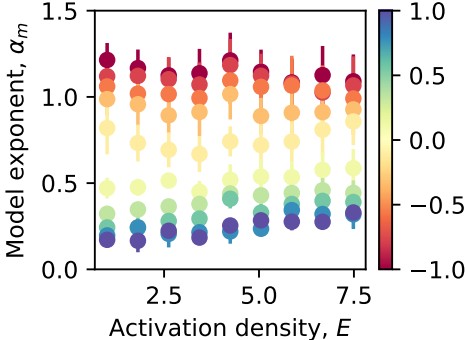

Figure 15: Model exponent is robust to activation density at different levels of superposition. The colorbar encodes weight decay as in the main text. Error bars are standard errors.

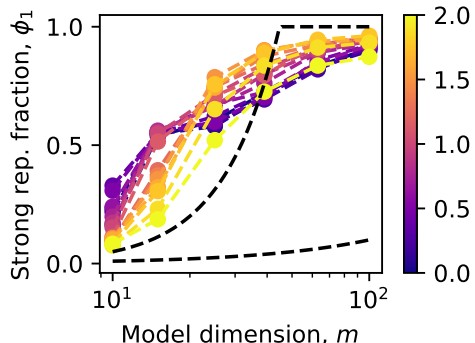

Figure 16: Fraction of strongly represented features ($\|W_i\|_2 > 1$) at strong superposition ($\gamma = -1$) is around $\min\{1, m^2/2n\}$ (fast increasing dashed line), which is much larger than $m$ (slowly increasing dashed line). Colorbar means $\alpha$ as the main text.

### D.5   Figure 4

In Figure 4a, we plot the raw losses from small toy model experiments (hyperparameters in Appendix B.2). The theoretical value of loss is approximated by an integral $\int_{n\phi_{1/2}}^{n} 1/i^\alpha \mathrm{d}i$, which is

$$
L = \begin{cases}
\dfrac{\phi_{1/2}^{1-\alpha} - 1}{1 - n^{1-\alpha}} n^{1-\alpha}, \ \alpha \neq 1, \\[4mm]
-\dfrac{\ln \phi_{1/2}}{\ln n}, \ \alpha = 1.
\end{cases}
\tag{14}
$$

To quantify how much the learned weight matrix deviates from the ideal no superposition structure, we construct a reference matrix and compute a norm difference. Specifically, we first create an $n$-by-$n$ zero matrix called base, and then insert an identity matrix of size $m$ in its top-left corner. This padded identity matrix serves as a reference for the perfect recovery of the first $m$ features. We then compute the matrix product $WW^T$ from the learned weights and compare it to this reference using the matrix 2-norm. The resulting value reflects the ambiguity or interference in the learned representations. We store this norm in the ambiguity tensor at the location indexed by the current task and model width. Given a weight decay and data exponent, we have 6 ambiguity values since we have 6 $m$ values. We calculate maximum ambiguity among these 6 models, and choose the 9 cases with the smallest maximum ambiguity to plot in Figure 4b. One can see that when weight decay is near $0.5$, the models are closest to the ideal no superposition case where the first $m$ features are represented perfectly. Smaller weight decay may not be sufficient to eliminate superposition, and larger weight decay can suppress features that, in principle, can be represented perfectly.

### D.6   Figure 5

For convenience, based on the observation that the vector norms are bimodal around 1 in the strong superposition regime, we define strongly represented features as those that have $\|W_i\|_2 > 1$, which are more frequent and ETF-like, and weakly represented ones for those with $\|W_i\|_2 < 1$. We can quantify the fraction of strongly represented as

$$
\phi_1 = |\{i : \|W_i\|_2 > 1\}|/n,
\tag{15}
$$

which significantly exceeds $m/n$ and is around the ETF upper bound $m^2/2n$ (Figure 16). The group of vectors $\|W_i\|_2$ with norm greater than 1 or the strongly represented features then roughly agree with ETF properties: small variance, $1/m$ mean squared overlaps, and a limited number of vectors.

Figure 5 studies the results from small toy models (Appendix B.2) focusing on the strong superposition regime. For the strongly represented fraction, $\phi_1$, we can directly compute based on the definition and the obtained weight matrices. We showed a row norm distribution at $m = 15$, $\alpha = 1$, and $\gamma = -0.78$ in Figure 5 panels a and b. Here, we provide more data to show that 1 is a natural separation point in norm to determine which are strongly represented and which are weakly represented (Figure 12).

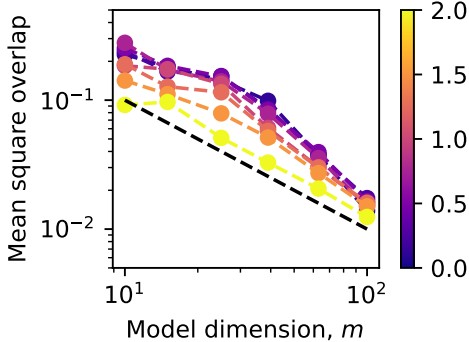

Figure 17: The mean squared overlap over all the $W_i$ vectors in the strong superposition regime ($\gamma = -1$). Although the value may be higher than $1/m$ (the dashed line), the scaling is robustly $1/m$. Colorbar means $\alpha$ as the main text.

Once select the rows with norm greater than 1, we can calculate their mean and variance of squared overlaps based on normalized rows $W_i/\|W_i\|_2$ (Figure 5, c and d). We argue that after training, the vectors will be more similar to ETFs than to random initialization. This is studied via the variance of overlaps. ETFs, in theory, have zero variance. We find that the majority of the overlap variances are much smaller than the random initialization, especially when features have similar frequencies, which agrees with the expectation. The cases where the actual variance is greater than that of the random vectors have large $\alpha$, roughly correspond to the cases where $\alpha_m$ deviates from 1 — ETF-like configuration no longer dominates. This is intuitive that when $\alpha$ is too large, the heterogeneity of overlaps will become large — it is better to let more frequent features occupy larger angle space. We argue that the large variance at large $\alpha$ does not mean the configuration tends to be random, but tends to be something more closely related to the frequency distribution of the features.

Our explanations based on the strongly and weakly represented features capture the basic trend that when $\alpha$ is getting large, the more important features will have larger angle space and the loss decay will be more related to the data exponent. However, this theory is oversimplified, where the strongly represented all have small overlaps and the weakly represented all have large overlaps. The real situation may be more like the angle occupied by one feature decreases continuously as the frequency decreases. As suggested by Figure 5c, overlap variance within the strongly represented is greater when $\alpha$ is larger. To be more precise about the overlap distribution as well as the exponent $\alpha_m$ when $\alpha$ is large, we cannot use simple theoretical expectations like ETFs but have to solve the toy model.

We also provide evidence that overlaps of all the vectors (Figure 17). We see that some of the mean square overlaps are larger than $1/m$ instead of being on the line $1/m$. However, all the mean values follow $1/m$ scaling even for large $\alpha$ cases where the vectors are no longer isotropic. We emphasize that for even frequencies and isotropic vectors, since squared overlaps scale as $1/m$, the loss should scale as $1/m$.

After fitting $\alpha_m$ of the trained small toy models (Appendix B.2), we plotted the $\alpha_m$ corresponding to the second to the fourth smallest weight decays in Figure 5e. We also copied from Figure 4b and plotted the ideal weak superposition case in Figure 5e. One question we had is that if $m^2/2$ is always greater than $n$, in our analysis, all vectors can be strongly represented, what should $\alpha_m$ be? We trained the small toy models again as in Appendix B.2 but with $m$ from 50 to 150. We found that in the strong superposition regime, the $\alpha_m$ is still around 1 when $\alpha$ is smaller than 1.5, and $\alpha_m$ still increases a little while smaller than $2(\alpha-1)$ when $\alpha$ is larger than 1.5 (Figure 18). When $m^2/2n > 1$ is always true, the vectors can be put into a configuration where all overlaps are small and scale as $\sqrt{1/m}$, such that $\alpha_m$ should be closer to 1. However, as mentioned before, our picture that the strongly represented have nearly uniform absolute overlaps is oversimplified. In the real situation, more frequent features have smaller or even faster decaying overlaps. Therefore, when $\alpha$ is too large, a weighted sum of squared overlaps, weighting the more frequent features more, can decrease faster than the average decaying speed $1/m$. Again, we need to solve the toy model faithfully to uncover the rigorous relation between $\alpha_m$ and $\alpha$ and argue the robustness of $\alpha_m$ from theory.

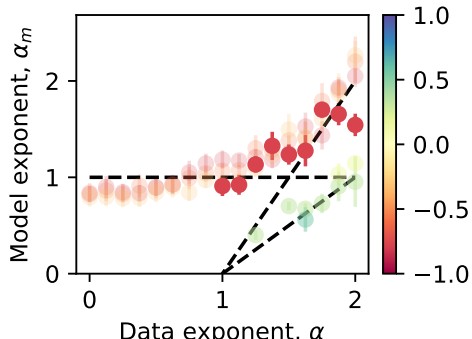

Figure 18: Small toy models with $m$ from 50 to 150 (such that $m^2/2 > n$) in the strong superposition regime yield similar $\alpha_m$ around 1 when $\alpha$ is small and a slightly smaller $\alpha_m$ (smaller than $2(\alpha - 1)$) when $\alpha$ is large. We copied Figure 5e and made the points transparent for comparison. The non-transparent points are from small toy models with $m$ from 50 to 150. $\alpha_m = 1$ is the horizontal line, $\alpha_m = 2(\alpha - 1)$ is the fast increasing dashed line, and $\alpha_m = \alpha - 1$ is the slowly increasing dashed line. Error bars are standard errors. The colorbar encodes weight decay as in the main text.

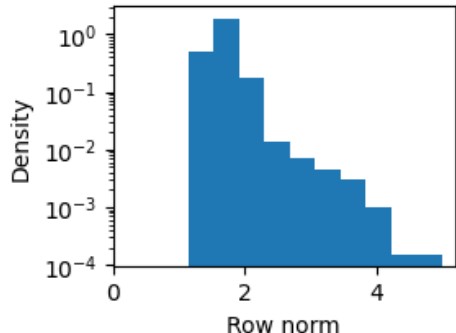

Figure 19: Row norm distribution of the language model head of OPT-125M.

### D.7 Figure 6

After obtaining the overlaps as described in Appendix C.1, we directly plot the raw data in Figure 6a. The data are quite noisy, and we did not fit the data with a line.

We argued that the LLMs are in the strong superposition regime since all tokens are represented. Figure 19 shows a typical row norm distribution of LLM (opt, 125M parameters [39]). We showed the mean, minimum, and maximum row norms of all the LLMs studied in Figure 20. From the non-zero minimum norms and the fact $n \gg m$, we confirm LLMs are in strong superposition. As mentioned in the analysis in Appendix A.2, we argue that the row norm of LLMs should not depend on $m$ but controlled more by the intrinsic data property of language, which is also verified to be valid (Figure 20).

We obtain the evaluation loss of each model on each dataset as described in Appendix C.2. We fit our loss values by the formula,

$$L = C_m/m^{\alpha_m} + L_{\backslash m},$$

where $C_m/m^{\alpha_m}$ is universal and $L_{\backslash m}$ is a constant depending on the dataset and model class. There are in total 16 different $L_{\backslash m}$ since we have 4 different model classes and 4 datasets. In our fitting model, there are in total 18 parameters. We use Adam to minimize the mean square error between the predicted loss by the above function and the real loss. All losses obtained are used in optimization. The code is in `nonlinearfit-3.ipynb`.

We provide the raw data, losses, as a function of $1/m$ (Figure 21). The losses looks like a line with $1/m$ in one model class and with the same dataset. And the slope of the line seems to be universal.

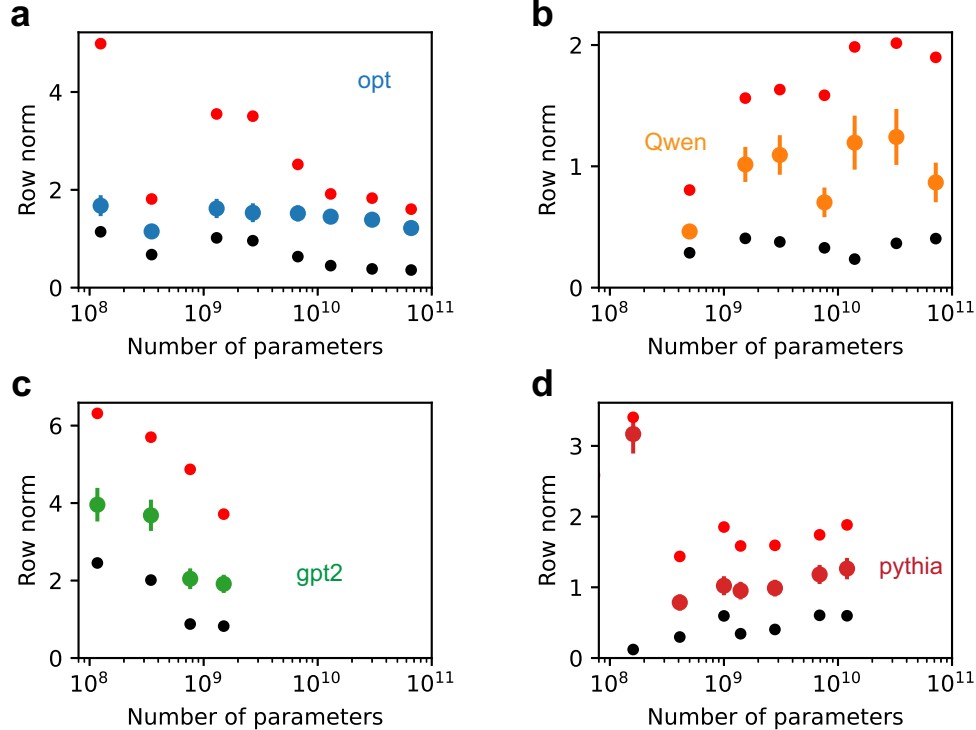

Figure 20: LLMs are in strong superposition based on the non-zero norms of the representation vectors. (a) OPT models. (b) Qwen2.5 models. (c) GPT-2 models. (d) Pythia models. The dots with error bars are mean values, and the error bar is the standard deviation of norms. Red small points are the maximum values, and dark small points are the minimum values. The mean norm value as a characteristic row norm does not depend on model size much as expected (Appendix A.2).

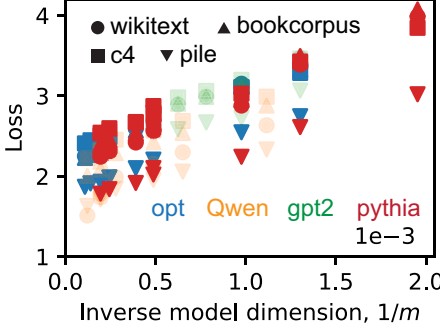

Figure 21: Raw evaluation losses as a function of inverse model dimension.

These two points support us in proposing the formula above, where $C_m/m^{\alpha_m}$ is universal. The intersections are different depending on the dataset and model class, corresponding to different $L_{\backslash m}$.

We obtained the token frequencies as described in Appendix C.3. Given the raw data, we sort the token frequency and obtain the frequency-rank plot. We sample 1000 (this number does not matter once it is large, 10000 gives the same result) points uniformly in the $\log_{10}(\text{Rank})$, and fit the frequency-rank as a power law, or a line in log-log plot. Results show that the data exponent fitted $\alpha$ is close to 1 regardless of the dataset or the tokenizer (Figure 22).

We study the relationship between model dimension $m$ and model size $N$ (number of parameters). For the four open-sourced models we analyzed [39–42], we can see that $N \sim m^3$, especially when $m$ is large. If we fit the $N$-$m$ relation by a power law while assuming a universal exponent but different

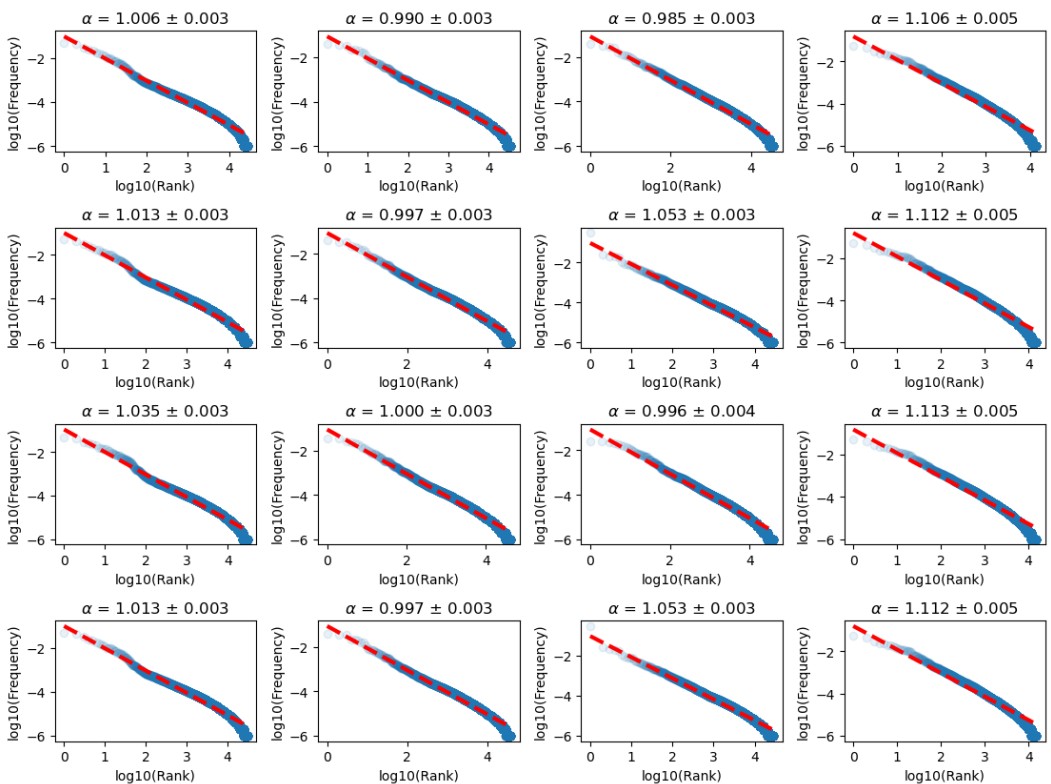

Figure 22: Taking tokens as atomic features, their frequencies indeed follow a power law, and the measured data exponent $\alpha$ is close to 1. The four rows from top to bottom correspond to four tokenziers, Pythia, OPT, Qwen2.5, and GPT-2, respectively. And the four columns from left to right correspond to four datasets analyzed, wikitext, C4, the Pile, and Bookcorpus, respectively.

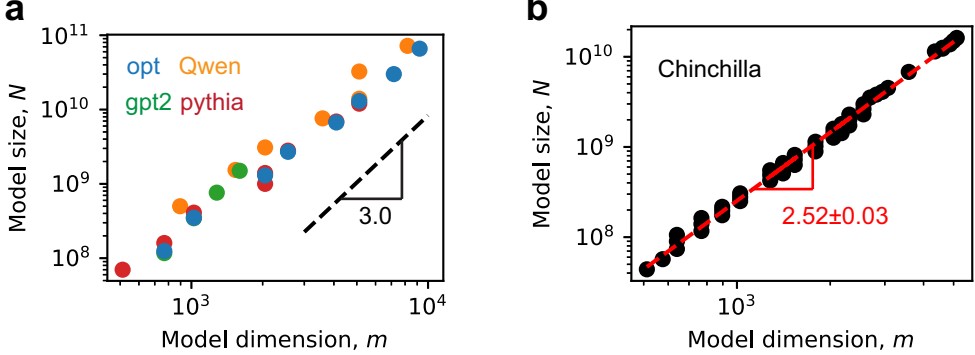

Figure 23: The model size is approximately a power law with model dimension. (a) The four model classes we analyzed [39–42]. (b) The Chinchilla models [3].

coefficients depending on the model class, we obtain an exponent of $2.51$ (Figure 23a). For the Chinchilla model reported in [3], we find $N$ is also close to a power law with the model dimension, and the fitted exponent is $2.52 \pm 0.03$ (Figure 23b).

