# OpenReview forum: "Superposition Yields Robust Neural Scaling"
_NeurIPS.cc/2025/Conference — NeurIPS 2025 oral_

### Official Review · Reviewer_xNH3 · 2025-06-10

**Clarity:** 3
**Significance:** 3
**Originality:** 3
**Rating:** 5
**Confidence:** 2

**Summary:**

This paper explores why larger language models (LLMs) perform better by examining the role of representation superposition in neural scaling laws. The authors construct a toy model to study how superposition and data structure influence the scaling of loss with model size. They find that under strong superposition, where all features are represented but overlap, the loss scales robustly as an inverse power law with model dimension, largely independent of feature frequency distributions. In contrast, weak superposition shows more fragile scaling behavior. The study concludes that representation superposition is a key mechanism underlying observed neural scaling laws and suggests that promoting superposition could lead to more efficient training strategies and model architectures for LLMs.

**Questions:**

In my understanding, a larger dimension implies greater superposition, which also means more parameters. How can we be inspired to achieve better performance with fewer parameters?

**Ethical Concerns:**

["NO or VERY MINOR ethics concerns only"]

**Limitations:**

The study focuses primarily on representation loss and does not explore other factors like dataset size or training steps in detail.

**Quality:**

3

**Strengths And Weaknesses:**

Strengths:

(1) The paper provides a comprehensive analysis of how representation superposition affects neural scaling laws, offering new insights into why larger language models perform better.

(2) It systematically studies the relationship between superposition and data structure, using both theoretical analysis and extensive experiments to support its conclusions.

(3) The findings have practical implications for developing more efficient training strategies and model architectures, potentially leading to better performance with less computation.

Weakness:

(1) The paper relies on a simplified toy model, which may not fully capture the complexities of real-world large language models.

(2) The theoretical analysis is not fully rigorous, leaving some deeper questions unanswered and requiring further investigation.

(3) The study focuses primarily on representation loss and does not explore other factors like dataset size or training steps in detail.

---

> ### Author Rebuttal · Authors · 2025-07-29
>
> We thank the reviewer for the nice comments and helpful questions. We will reply to the weaknesses and limitations point by point.
>
> * Weakness 1: We agree, and we may emphasize this weakness more in our discussion of limitations. We tried to use the last paragraph in Section 2 to summarize some key differences between the LLMs and the toy model and justify why some details in LLMs can be ignored, which should also be highlighted or strengthened.
>
> * Weakness 2: This is true. The robustness of $\alpha_m\approx 1$ across $\alpha$ and the transition of $\alpha_m$ are observations, and we try to provide a picture of the underlying mechanisms with more details like vector norms and overlaps observed from the experiments. However, we are limited by the fact that we are not able to analytically solve the directions of the representation vectors. We are actively trying many theoretical analysis methods. One simplified analysis is to assume random directions of the embedding vectors and solve for norms and biases.
>
> * Weakness 3: Thank you for mentioning this. We had a brief discussion on other scaling laws in Appendix B. We are now actively studying the scaling with training steps. The scaling with dataset size should be the same as that with training steps in the online learning setup. The training dynamics in the weak superposition regime have been solved. Yet, the training dynamics in the strong superposition regime is hard to analyze.
>
> * Questions: Given the number of features, $n$, a larger dimension $m$ may lead to less superposition. When we talked about equal performance with fewer parameters, we were imagining a large model with weak superposition and a small model with strong superposition could have similar losses (which is true in the toy models).
>
> * Limitations: Please see reply to Weakness 3.

---

> > ### Comment · Reviewer_xNH3 · 2025-08-02
> >
> > I have reviewed your responses and the revisions made to the manuscript. Most of the concerns and my initial questions have been adequately addressed. However, one remaining issue needs clarification:
> >
> > Could you please provide a justification for the assumption of random directions of the embedding vectors?‌
> >
> > ‌Aside from this point, I maintain my original evaluation score for the paper.

---

> > > ### Author Response · Authors · 2025-08-04
> > >
> > > Thank you. Our mention of random embedding vectors is intended to illustrate one approach for deeper theoretical analysis (not just scaling but also the loss values and transition in scaling).
> > >
> > > The random vectors can be modeled as quenched disorder in physics literature and have been studied in other fields such as  compressed sensing. And in terms of performance, the random vectors have expected overlap being $1/\sqrt{m}$, which is close to ETF when $n\gg m$. Therefore, random vectors are an analyzable ansartz that may provide theoretical insights for cases where $n \gg m$.
> > >
> > > We believe there are many more things to do on the theory side, and random vectors are a good starting point.

---

### Official Review · Reviewer_hFfh · 2025-07-02

**Clarity:** 3
**Significance:** 3
**Originality:** 3
**Rating:** 4
**Confidence:** 3

**Summary:**

This work analyzes neural scaling laws in terms of the concept of superposition. The main body of the paper centers around a toy model for superposition first advanced by a team at Anthropic. In this model a small two-layer network is trained on a synthetic dataset where all features in the dataset can be directly controlled. The work identifies an optimization scheme such that strong superposition and weak supervision can be controlled via a weight decay parameter. Here, strong supervision is the state where the hidden activations of features fail to be fully orthogonal and the extent to which they are not orthogonal is determined by the frequency by which the feature appears. Weak superposition on the other hand is the situation where a subset of the features are represented by orthogonal vectors in the model and the rest are mapped to zero. The paper looks at how adjusting this hyperparameter, as well as the distribution of features impacts scaling constants in the model. They find two regimes. One, corresponding to strong superposition, where scaling closely follows a power law and one where it fails to follow a power law corresponding to weak superposition. The paper justifies this through an analysis of their toy model. The paper goes on to explore the statistics of large-scale LLMs and provides evidence that these fall into the strong superposition scheme.

**Questions:**

- The paper changes the hyperparameter settings so as to induce different kinds of superposition. The paper then observes different scaling properties. There is certainly a correlation here, but how do we know that superposition causes the scaling laws (it is possible that this falls out of the analysis and I am misremembering)?
- It is this reviewer’s understanding that the paper does not consider the case where features are correlated. This is an essential feature of real-world data and it would be interesting to consider this. Have the authors thought about how this would change their model?
- What does the statement ‘Conceptually, fitting…’ on Line 46 mean?

**Ethical Concerns:**

["NO or VERY MINOR ethics concerns only"]

**Final Justification:**

The reviewer believes that this paper represents a promising direction of research with some interesting initial results. The primary aspects holding back the paper are presentation issues and lack of connection to real-world examples. The reviewer believes that these are both very addressable and hopes to see a new version of this work soon.

**Limitations:**

The paper does not address limitations. Limitations might include the fact that most of the analysis is done on a very simple model and would need to be supported by evidence in larger, more real-world datasets/models.

**Paper Formatting Concerns:**

I have no formatting concerns.

**Quality:**

3

**Strengths And Weaknesses:**

Strengths:
- **Analyzing the relationship between scaling laws and superposition is a natural research direction:** Scaling laws and superposition are both central ideas that have arisen through the development and study of LLMs. It is natural to ask in what ways they may be related. The paper lays out a nice framework with which to think about this.
- **The analysis of the toy set-up is comprehensive and yields a nice story:** Though the toy model used in this paper is not a contribution of the work, the paper puts it to good use and it is a very helpful aid when reasoning through the ideas presented and discussed by the paper. The notation and descriptions of the experiments are reasonably easy to follow and do not ask too much of the reader’s mathematical background.

Weaknesses
- **The structure of the paper could be improved:** The paper is primarily one long analysis of the toy model. This analysis could easily be partitioned into sections that would help organize the content. For instance, the analysis of the weak superposition regime could be a section (or subsection), the analysis of the strong superposition regime could be a section, and the analysis of non-toy models could be a section. This would enable the main takeaways of the paper to be more easily extracted and would help readers that do not want to read through the paper linearly. It would also be helpful to provide a conclusion or discussion section which summarizes the results and puts them into context.
- **The section on real-world models is weak when compared to the analysis of the toy set-up:** As noted above, the reviewer felt that the toy dataset section was helpful and made the paper much more interesting since it enabled a more detailed analysis both from the perspective of being able to finely control the whole system for experiments (including the data distribution) but also being able to theoretically justify various observations. However, this toy set-up is very far from real machine learning. The paper includes a short section in the end where real LLMs are analyzed for some of the same properties that were observed in the toy model, but the reviewer found this far less convincing than the rest of the paper. It is unreasonable to ask too much in this larger setting, but a more detailed analysis would be helpful. Furthermore, there may be small scale experiments that can be run that still use real data and our easier to analyze than large-scale LLMs.

Nitpicks
- The paper uses the terminology ‘weak representations’ and ‘weak superposition’, the reviewer would suggest changing the former as these are not the same thing and may be confused with the current naming convention.
- Line 40: ‘depend sensitively on’ $\mapsto$ ‘are sensitive to’.

---

> ### Author Rebuttal · Authors · 2025-07-29
>
> We thank the reviewer for the helpful questions and suggestions. We will reply to the weaknesses, questions, and limitations point by point as follows.
>
> * Weakness 1 (Structure of the paper): We are sorry for the confusion and inconvenience due to our presentation. The reviewer's suggestion is very helpful, which we will follow. We will also highlight our key definitions, assumptions, and takeaways by the Theorem environment. We attempted to include too many results in the main text. We will emphasize only the most important ones and save space to move Discussion back to the main text.
>
> * Weakness 2 (Weak LLM analysis): Yes, we totally agree with the reviewer, and we should acknowledge this limitation more. Our toy setup is far from actual LLMs, and we used the last paragraph of Section 2 to discuss key differences and why some details in LLMs may not matter. In Discussion (Appendix B), we also discussed other sources of error in LLMs and how to better interpret Figure 8b. Nevertheless, the connection between toy models and LLMs is not rigorous. Regarding the analysis of LLMs, we studied the embedding vectors, loss scaling with width, and token frequency distribution (Figure 22). We tried to analyze properties we have access to and that can be connected to the toy model in some way. The philosophy is that we study all kinds of possible behaviors in the toy model, identify the regime where LLMs are in, and connect LLM behaviors to mechanisms found via the toy model. We can move more details on LLM analysis from the Appendix to the main text to strengthen the last part. We are also willing to conduct mechanistic investigations on LLMs if there is any concrete quantity to look at.
>
> * Nitpick 1: Thank you for this point. We can use "well represented" vs. "poorly represented" or "over represented" vs. "under represented".
>
> * Nitpick 2: Thanks, we will revise.
>
> * Question 1 (Causality?): This is a conceptually important question. Let's think about another example first. Say we can change the temperature to make water ice or liquid. A behavior we found at the lower temperature state is a behavior of the ice. And a behavior at higher temperatures is that of the liquid state. By changing weight decay, we enter the state of "weak superposition" or "strong superposition". We can always say that the scaling law is a property of the strong superposition regime. To be rigorous, "geometry of representations in the strong superposition regime yields a scaling law with width" is the statement, and "superposition causes scaling law" is a simpler version of it.
>
> * Question 2 (Correlation of features): This is a great question. If we consider the correlation between different dimensions of the input vector, the "independent atomic features" should be the eigenvectors of the correlation matrix instead of the dimensions of the input vector. Theoretically, after rotating the weight matrices to another basis, the current analysis still follows. Yet, the non-linearity can introduce extra complexity, which we are not certain about and should study via experiments. We expect the scaling to be robust even if input dimensions are correlated. We should acknowledge this point in our Discussion. And if we consider the eigenvectors of the correlation matrix to be features, the feature frequencies can be different from those of the input dimensions. We measured the co-occurrence matrix of tokens in the Pile, varying context length from 3 to 512. The eigenvalues of the co-occurrence matrix have a scaling exponent $\alpha$ from 1 to 1.2, which does not change much. So, we also do not expect that considering token correlation would change our conclusion.
>
> * Question 3 (Meaning of a sentence): We are sorry for the confusion, "fitting function or manifold" and "learning skills" refer to previous mechanisms proposed to explain the neural scaling laws. We will add citations near these terms. The sentence tries to say that some previous mechanisms conceptually describe what the transformer layers do, while there is another source of error -- representation or embedding -- that needs studying.
>
> * Limitations: We discussed some of our limitations in Appendix B Discussion. We will emphasize what the reviewer pointed out. And we will try to move some results to the Appendix such that we can move the Discussion section back to the main text.

---

> > ### Comment · Reviewer_hFfh · 2025-08-04
> > **Reply to authors**
> >
> > The reviewer would like to thank the authors for their clarifications. We would also like to encourage the authors to keep at this paper. We believe it has interesting ideas and results but needs some presentation revisions to better communicate these. We will keep our current rating.

---

### Official Review · Reviewer_H5Wk · 2025-07-03

**Clarity:** 4
**Significance:** 4
**Originality:** 4
**Rating:** 6
**Confidence:** 3

**Summary:**

This paper proposes an explanation for the power-law scaling behavior of LLMs' loss with model size, connecting it to superposition—the phenomenon where neural networks represent more features than their model dimensions by allowing some degree of interference (non-orthogonality) between feature representations. The authors build upon a previously introduced single-hidden-layer toy model, originally designed to demonstrate superposition, modifying it by incorporating a weight decay (or growth) term into the loss. This modification enables control over the degree of superposition, allowing its effects to be studied independently of data properties. The authors find that in a weak superposition regime, where only the most important (frequent) features are represented without interference, loss scales as a power law with model size only if the data distribution itself follows a power law. However, in a strong superposition regime, where all features are represented with interference and this interference dominates the loss, the loss scales as a power law with model size regardless of the data distribution. They provide a geometric explanation for this behavior in their toy model: in the strong superposition regime, the optimal representational strategy to minimize interference approximates an equal-angle tight frame. Under these conditions, the squared dot product between representations scales inversely with model size according to a power law with exponent 1. The authors then analyze the unembedding layers of several publicly available LLMs within a framework analogous to their toy model, finding evidence that these models operate in the strong superposition regime. This suggests that the mechanism identified in the toy model might also dominate the loss behavior of real LLMs, potentially explaining the empirically observed power-law scaling with model size. Furthermore, the empirically observed scaling exponents closely match the predicted value of 1. The authors also demonstrate alignment with the Chinchilla scaling laws by converting Chinchilla’s parameter-count-based term into a model-width-based term (consistent with their single-hidden-layer framework), again obtaining an exponent close to 1.

**Questions:**

The points below are intended for discussion—feel free to consider them low priority and/or disregard some of them. The questions are intuition-driven, so please forgive any lack of rigor.

- I would like to better understand your thinking behind the following phrasing:

  > *"Conceptually, fitting functions or manifolds and learning skills or grammars are primarily tasks of the transformer layers, while representation learning is more directly tied to the embedding matrix and the language model head."*

  Defining representation learning strictly as learning token embeddings seems unusual to me. Aren’t intermediate activations also representations, capturing increasingly abstract and composed features (at least up to the middle layers)? Would it perhaps be beneficial to introduce a more specific term for what you call "representation loss," clarifying that you specifically refer to non-contextual representations?

- Given the compositional process across layers, which yields a combinatorially increasing number of features far exceeding the vocabulary size, should we expect loss due to superposition (geometric interference) to play an even more dominant role in what you've termed the _parsing loss_? Would this make weak superposition virtually impossible in LLMs?

- If we do not assume the Linear Representation Hypothesis to hold, would this have any direct implications for your theory? In its current form I guess it would not, given that the non-contextual representations (token embeddings) you analyze are linear by definition, but I'm curious whether you foresee any implications arising from such a scenario—perhaps specifically for the parsing loss.

- Since natural language data often follow power-law distributions, other theories relying on this assumption should also be considered plausible explanations for LLM scaling behavior. It would be beneficial to see a discussion regarding how compatible or complementary your theory of neural scaling with model size is relative to previously proposed theories.

- What do you consider the most likely approach for falsifying your theory?

- Considering that superposition has mostly been presented as a negative phenomenon and a significant interpretability challenge for neural networks, I suggest discussing the potentially adverse implications of your proposal to encourage superposition as a strategy for improving performance. Especially given that AI safety is becoming increasingly prominent, this concern merits acknowledgment and thoughtful discussion.

- Typo in the caption of Figure 6: "at stong superposition" → "at strong superposition".

**Ethical Concerns:**

["NO or VERY MINOR ethics concerns only"]

**Final Justification:**

I will maintain my score. I found the paper to be technically solid, innovative, and potentially of significant impact in the field, both in advancing understanding and in driving improvement. Most of the weaknesses identified by other reviewers appear to stem from difficulties in reading the paper, which I believe will be addressed in the final version.

**Limitations:**

The authors should discuss the potential negative implications for AI interpretability and safety arising from their suggestion to promote superposition in neural networks as a means of improving performance.

**Quality:**

4

**Strengths And Weaknesses:**

**Strengths**:

- A compelling and somewhat intuitive argument presenting a novel connection between two highly relevant and interesting neural network phenomena: superposition and scaling laws.
- A useful and straightforward extension of the superposition toy model that allows interventions to induce or inhibit superposition, resulting in observations largely independent of data distribution or model size. This differentiates it from previous explanations of neural scaling laws that relied on data-distribution assumptions (e.g. uniform on a d-dimensional manifold [15]; or power-law distributed quanta [19]).
- A rigorous, experiment-rich investigation of the toy model, revealing interesting behaviors across varying data distributions and degrees of superposition, consistent with a geometric explanation of power-law scaling under optimal arrangement conditions to minimize interference in superposition.
- Insightful experiments with real LLMs that provide empirical support for the proposed theory.
- The work goes beyond a purely descriptive model, potentially offering actionable insights for improving LLM architectures and training strategies.
- The authors suggest many different directions for future work based on their model and gained insights, showcasing that it can be an important contribution to the community even beyond the results presented here.
- Clearly written and overall very well presented.

**Weaknesses**:

Any potential limitation I identified has already been acknowledged and thoroughly discussed by the authors and does not, in my opinion, remotely warrant rejection. See the Questions section for additional points intended for discussion.

---

> ### Author Rebuttal · Authors · 2025-07-29
>
> We are grateful for the reviewer's appreciation and helpful questions. We will reply to the Questions and Limitations point by point as follows.
>
> * Question 1 (Phrasing of a sentence): We are sorry that we misused the terminology "representation learning". Our representation-limited error is due to the imperfect token or concept embedding in a low-dimensional embedding space. In language, tokens are atomic features in our mind, and we can conceptually connect them to features in our toy model. And practically, token embeddings are easy to access and analyze. Yes, if the token embedding has some error, the more abstract representations in the middle layers will also have an error. We guess the error of composite features would have the same scaling with width as the atomic features. Our thinking behind the sentence is that there can be two sources of the total error, one is from the imperfect embedding itself (explicit focus of this paper), and the other is from imperfect calculation or transition acting on the embeddings (which is more related to the transformer layers). We propose keeping the phrase "representation" as it could apply to either components of error, although we could also replace it with "embedding" if this would avoid confusion.
>
> * Question 2 (Middle layer superposition): This is a great question! In the AI biology work by Anthropic, they found that in poem completion tasks, the model can plan ahead by representing the next whole sentence at the end of the prompt using the embedding dimension of one token. In the middle layers, the number of concepts to represent may be huge, especially when the context length is large. So, in our mind, for LLMs, superposition may be impossible to avoid. Yet, we do not have more rigorous arguments for this. Token embedding is a subproblem for representation in LLMs and should share the same scaling. There is definitely more to study in representations in the middle layers, and we can add discussions on this topic in the Discussion section. We also imagine that, in terms of new architectures, an adaptive hidden dimension may help performance: if one position needs a lot of previous tokens to understand or is needed by a lot of following tokens for their decoding, it should have a larger embedding dimension.
>
> * Question 3 (Linear representation hypothesis): Thanks for this question. If the linear representation hypothesis (LRH) holds, we expect the parsing loss to have a term that is the number of compositions times the error of one embedding. If LRH does not hold, there are a lot of possible situations. For example, the embedding may be on a manifold. But as long as the manifold is locally Euclidean, the 1/width scaling may still be robust (we are not sure). Another possibility when LRH does not hold is that there are infinitely many independent features, and LLMs need to keep embedding new independent features in the middle layers, which may increase the coefficient but not change the 1/width scaling. We are also sorry for the lack of rigor and preliminary thinking.
>
> * Question 4 (Previous theory): Thanks for mentioning this. Since different theories on neural scaling laws have different setups, we are not very confident in comparing them directly. As far as we can see, previous theories like the phenomenological quanta model and the formal kernel method have the same spirit as our no superposition result. Our finding in the strong superposition regime is therefore complementary to previous theories, suggesting a new mechanism due to the intrinsic model configuration rather than data properties.
>
> * Question 5 (Falsifying our theory): We think a direct experiment is to fix a large depth and only increase the width of LLMs, and study clearly the scaling of loss with width. If the scaling exponent obtained from this experiment is far from 1, then our mechanism cannot explain LLM behaviors. Another approach is about language itself. If one can argue that atomic features of language are not tokens but something much fewer than the number of tokens and embedding dimension, then our theory is false.
>
> * Question 6 (Interpretability and AI safety): Thank you for reminding us about interpretability and safety issues. We agree that encouraging superposition may lead to more powerful but less interpretable models and can have a negative impact on safety. We will add a discussion on these topics.
>
> * Question 7 (Typo): Thank you for the careful reading. We will revise.
>
> * Limitations: Please see reply to Question 6.

---

> > ### Comment · Reviewer_H5Wk · 2025-08-02
> >
> > Thank you for the thoughtful responses. After reading your rebuttal to my review and those of the other reviewers, I have decided to maintain my score.
> >
> > Regarding the potential change from “representation” to “embedding,” I believe it would be an improvement. However, since no other reviewer has raised this point, I’m fine with you keeping the “representation” phrasing if you prefer.
> >
> > I’ll put some more thought on your responses and hope to have the opportunity to discuss them further at the conference.

---

> > > ### Author Response · Authors · 2025-08-04
> > > **Reply to reviewer comment**
> > >
> > > Thank you for the further comments.
> > >
> > > We shall clearly explain that in our quantitative analysis, we studied token embedding, yet we believe or hope that our result holds for a more general representation of other things. We will add discussions on this and other topics the reviewer mentioned.

---

### Official Review · Reviewer_Y6XA · 2025-07-07

**Clarity:** 3
**Significance:** 3
**Originality:** 4
**Rating:** 5
**Confidence:** 4

**Summary:**

The paper argues that superposition is an important mechanism underlying observed neural scaling laws. To show this, the authors consider a toy setting adapted from the original toy models of superposition work and study how the level of superposition, controlled by a specifically defined weight decay term, influences scaling behavior. The key finding is that under weak superposition, power-law scaling occurs only if the features themselves are power-law distributed (model exponent is $\alpha_m \approx \alpha - 1$, where $\alpha$ is a data exponent), while under strong superposition, power-law scaling emerges largely independent of the feature distribution (model exponent, however, still depends on the data exponent, following $\alpha_m \approx \max(1,\ 2(\alpha - 1))$). Beyond the toy setting, the paper also analyzes token representations in the last-layer weight matrix of practical LLMs and shows that they exhibit power-law scaling with a model exponent $\alpha_m \approx 1$, consistent with the toy example in the strong superposition regime with $\alpha \approx 1$. This connection is then further extended and tied back to the scaling behavior of the cross-entropy loss in LLMs.

**Questions:**

All questions and concerns are detailed in the Strengths and Weaknesses section. The main weaknesses 2-5 will have the greatest impact on my final evaluation after the rebuttal.

**Ethical Concerns:**

["NO or VERY MINOR ethics concerns only"]

**Final Justification:**

**Initial submission.** The paper presented a novel and interesting view on scaling laws based on the idea of feature superposition. However, it had issues with text structure and clarity (Weaknesses 1, 2, 3), and some empirical results were not fully supportive of the claims (Weaknesses 4, 5).

**After discussion.** The authors clarified unclear points (Weaknesses 2, 3) and provided additional discussion on the claims that were not fully supported by the results, offering a reasonable explanation for the potential reasons behind the discrepancies between the model and the findings (Weaknesses 4, 5). They also promised to improve the text structure and incorporate the additional discussions in to the paper.

**Final recommendation.** The rebuttal and discussion addressed most of my concerns. While some questions remain open, the authors provided reasonable discussion on all of them. I believe that after incorporating these discussions and improving the structure of the text, the paper would make a strong contribution. Therefore, I now agree with other reviewers, will increase my score and recommend the paper for acceptance.

**Limitations:**

yes

**Paper Formatting Concerns:**

--

**Quality:**

3

**Strengths And Weaknesses:**

Strengths:
1. The paper addresses an important problem: understanding the underlying reasons behind neural network scaling laws is highly relevant in the current era of large models.
2. The idea of superposition and its role in scaling laws is interesting and, as far as I know, novel.
3. The paper focuses on a well-motivated toy example and makes a meaningful connection to practical models.
4. The authors make a commendable effort to provide a thorough analysis, linking theoretical and intuitive ideas with empirical evidence.

Main weaknesses:
1. The writing of the paper is very dense. In particular, the monolithic 4.5-page results section was difficult to read and navigate, and I found it challenging to precisely assess the claims due to how scattered they are throughout the text. I believe the paper would benefit from a significant rewrite to improve clarity and structure. Additionally, dedicating more space in the main text to discuss how the proposed explanation of scaling laws connects to existing theories (it is very important for placing the work in context) and comment on how this explanation could inspire new training methods or architectures (this point is mentioned in the abstract but discussed only in the appendix) would benefit the paper.
2. The choice of weight vector norms as indicators of features being represented should be more thoroughly explained, and their distributions along with the threshold values used should be commented on in more detail:
    - Initially, it was not clear to me what "feature being represented" means and that it is different from a feature being correctly predicted. Because of this, I was confused during the first two sections about the idea of learning more features than dimensions, since learning more features should directly harm the quality of how well each of them is learned. The second section of the paper should explicitly discuss that representing a feature is different from correctly predicting it, and that superposition is a balance between representing more features and having destructive interference between them during prediction.
    - The choice of the 0.5 threshold for defining whether a feature is represented or not is based on the empirical norm distribution shown in Figure 3a. However, as far as I understand, this distribution may vary significantly depending on the level of superposition (e.g., the norms are not close to one in the distribution shown in Figure 6b). Could you please clarify how the norm distribution changes across different superposition regimes, and whether the 0.5 threshold is meaningful in all cases, not only in the weak superposition setting?
    - Could you please comment on why the vector norms are distributed bimodally in the case of strong superposition shown in Figure 6b? Given the form of the weight decay term, one might expect the raw norms to be close to 1. Why is the density low near 1? Intuitively, since the feature frequencies change smoothly and monotonically, one might expect feature representation behavior to also change smoothly. Why do features instead fall into two distinct modes? Is this an artifact of the particular weight decay formulation? And if so, why should we expect this behavior to reflect what happens in practical LLMs?
3. I believe there is an important issue in the assumption made in equation (4). Specifically, $y_i$ are assumed to be equal to $0$ for features $i > m$. However, based on the original toy models of superposition work, for the unlearned features, $y_i = b_i$ where the learned bias term $b_i$ is the expected value of $x_i$.
4. The assumption that strong superposition is characterized by ETF-like geometry is not convincingly validated. The main evidence I gathered from the paper comes from Appendix F.6 and Figure 15, specifically the observation that the variance of feature overlap is smaller for strongly represented features than that of $m^2 / 2$ random vectors. However, this alone is not particularly convincing. Do weakly represented features behave differently? Is the number of strongly represented features actually equal to $m^2 / 2$ in practice? Not according to Figure 6c. In general, for such an important claim, I would expect a more rigorous and convincing analysis.
5. There is no clear explanation for the change in scaling observed with strong superposition as data exponents increase. The intuitive explanation that higher-frequency feature vectors become more separable (collapsing lower frequency vectors and increasing their errors due to feature overlap) is compelling, but seems to be inconsistent with Figures 6d and 16, where increasing data exponents actually increases feature overlap for strongly represented features and decreases feature overlap for weakly represented ones.

Additional concerns, comments, and questions:
1. Line 62: The weak superposition regime is not defined at this point.
2. Lines 88/95: It seems to me that “data” is usually used to refer to a dataset or data distribution, so it was a bit confusing that “input data” and “one data” referred to a single sample.
3. Figure 3a: The small subplots are difficult to parse and should be commented on in more detail.
4. Equation 2: Does this equation take into account the loss gradients or only the weight decay term?
5. Figure 4b: A diverging color map centered at $\alpha_m=1$ would benefit the figure. Currently, it is difficult to see that the values in the red box are close to 1.
6. Figure 4b caption: What does $m = 10 \sim 100$ mean here? Which model size is used?
7. Line 184: The word “nonlinearity” appears twice.
8. Figure 6c caption: “which is much larger than m (slowly increasing dashed line)” — I think it should be $m/n$.
9. Figure 6e: The second increasing line is not defined. Which weight decay values are used in these experiments? Where are the points for high weight decay values close to 1?
10. Why, in the analysis of token representations of LLMs, is the weight matrix of the last layer used rather than the embedding matrix? (Or are they tied in the considered models?)

---

> ### Author Rebuttal · Authors · 2025-07-29
>
> We thank the reviewer for the careful reading and professional technical questions. We reply to the questions and concerns point by point below.
>
> * Main weakness 1: We are sorry for the inconvenience due to our presentation. We wanted to include too many points in a limited space (9 pages), which led to dense writing and inadequate details for each point. To solve this problem, we plan to emphasize the core results in the main text by adding more details and moving less important results like Figure 7 to the Appendix. We can also add subsections and use the Theorem environment to clearly list our assumptions and results to improve readability. We are willing to incorporate more suggestions from the reviewers, if any.
>
> * Main weakness 2:
>     * Yes, features being represented are those having non-zero representation vectors. Features being well-represented means they can be correctly recovered or predicted. We can highlight this by using a "Definition" environment.
>     * Based on the definition of "represented or not", the threshold should be 0. In practice, we need to use a (small) non-zero threshold. We chose 0.5 since it is equally far away from 1 and 0, potentially minimizing misclassification. But this choice is very conservative since 0.5 is not small. When $\gamma \geq 0$, the norm distribution concentrates near 0 or 1 (Figure 11), and choosing a threshold from 0.05 to 0.95 does not change the outcome. When $\gamma < 0$, all norms are around 1 but may have large variance (Figure 12). In this case, choosing 0.5 as a threshold will give a conservative lower bound for the number of features being represented. Our goal is to establish a relation between $\gamma$ and superposition. If a lower bound is close to 1, it suffices to say that $\gamma<0$ can robustly generate strong superposition. We should add a remark on this issue. We can also provide a scan or sensitive study of the choice of thresholds in the Appendix.
>     * The number of ETF-like vectors has a bound $\sim m^2/2$. When $m^2/2<n$, the less important features cannot have small overlaps and have to decrease their norms to have less interference. The two distinct modes are being in ETF-like configuration (with mutually small overlap) or not. This particular behavior is irrelevant to LLMs since $m^2/2>n$ is always true. Please see Figure 19, LLM embedding vector norms is not bimodal. Then, LLM loss should be dominated by the strongly represented features (ETF-like), more likely leading to $\alpha_m = 1$, which seems to be true. We scan all kinds of behaviors of toy models, then identify which regime LLMs are in, and use the mechanism found from the toy model in that regime to explain the LLM behaviors.
>
> * Main weakness 3: We are sorry for our oversight. The assumption is that all biases are zero. In our experiments, since both $m$ and $n$ are large, the expected value of $x_i~(i>m)$, proportional to the probability of $p_i$, is extremely small, approximately satisfying the assumption.
>
> * Main weakness 4: "ETF-like" has two main points: (i) Overlaps distributed closely around $\kappa$ in Eq. (5); (ii) The number of such vectors has an upper bound. We verified point (i) by Figure 6d (i.e., the mean of overlaps agrees with $\kappa$) and Figure 15 (i.e., the variance is small at least for small $\alpha$). And yes, weakly represented features behave differently, as they have much larger mean overlap (compare Figure 16 and Figure 6d). We verified point (ii) in Figure 6c, where the bound is not exactly $m^2/2$ but around. This discrepancy is part of why we refer to the vectors as being "ETF-like" (we do not claim they form an ETF). What we can rigorously say is that when $\alpha = 0$, the optimal configuration is an ETF. However, when $\alpha$ increases, one would expect important features to occupy larger angular space and break the evenness. Yet, we still observe that the vectors are similar to ETF for a wide range of $\alpha$ and yield a robust $\alpha_m \approx 1$. In our Discussion, we admit our main weakness is that we cannot analytically solve the toy model. What we did was to connect empirical findings and simple theoretical examples to generate a zeroth-order understanding.
>
> * Main weakness 5: This problem is also due to the fact that we cannot solve the toy model now. Specifically, we found that solving the directions of the embedding vectors is unexpectedly hard. What we are actively trying is to assume random directions and solve vector norms and biases. We are curious how many phenomena can be explained by such an ansartz. What we presented in the paragraph starting the line 217 is a hypothesized explanation for the transition observed. Some results in Figure 16 are confusing to us, too. The proposed picture seems to agree more with observations in the small $\alpha$ regime, but not the regime after the transition. Our understanding is that when $\alpha$ is large, the overlap variance is also getting large (e.g., Figure 15). So, the decreasing mean overlap over weakly represented features does not mean they are not squeezed. The weakly represented ones may be squeezed into different positions in the space (and have lower mean but higher variance) such that the most important features can have lower overlap with each other. As long as each weakly represented feature contributes to constant loss (e.g., has one large overlap), our proposed explanation may be valid. Again, the study of direction configurations is hard in high-dimensional space for both theory and experiment. We should highlight that our explanation for the transition is one hypothesis. We believe that the empirical results are correct and worth sharing, and many theoretical studies can be done later by us or others with various tools.
>
> * Additional concerns 1: We defined "the absence of superposition" in line 59. We will rewrite to make it clear what "weak superposition" is.
>
> * Additional concerns 2: We will use "input sample" or "input".
>
> * Additional concerns 3: Thank you, we will explain more in the caption.
>
> * Additional concerns 4: Only weight decay. Since we used AdamW, weight decay is a separate line of code.
>
> * Additional concerns 5: Thanks for the suggestion! This would help a lot.
>
> * Additional concerns 6: $m = 10, 15, 39, 63, 100$. They are roughly uniform in the log scale. Will make this clear.
>
> * Additional concerns 7: Thanks! We will revise.
>
> * Additional concerns 8: You are right, we will revise.
>
> * Additional concerns 9: The second line is copied from Figure 5b. Therefore, we can summarize all behaviors of $\alpha_m$ in different regimes on one plot. We tested 10 $\gamma$ evenly from $-1$ to $1$. The points with high weight decay have $\alpha_m \ll 1$. And the reason is provided around line 166: if weight decay is too large, the number of learned features will be fewer than $m$, which leads to slow decay of loss and is not ideal.
>
> * Additional concerns 10: For models with sizes smaller than 3 billion, the embedding and LM head matrices are usually tied. Larger models usually do not tie the matrices. Analyzing either matrix leads to the same conclusion. However, analyzing the embedding matrix will lead us to a lengthy discussion on how the representation error would propagate through transformer layers, which may distract from the focus of this paper. We can provide the analysis result of the embedding matrices in the Appendix.

---

> > ### Comment · Reviewer_Y6XA · 2025-08-05
> >
> > Thanks for the detailed response!
> >
> > Most of my concerns are adequately addressed; however, I still have some **questions regarding Weaknesses 3 and 4**:
> >
> > **Weakness 3.** Why is this assumption necessary? Why not simply compute the loss with the correct mean values?
> >
> > **Weakness 4.** I am still confused about the second argument and Figure 6c. The empirical fraction of strongly represented vectors is significantly higher than $m^2/2$  for smaller $m$. Even for $\alpha = 0$, where the ETF should be the optimal configuration, the empirical results do not match the predicted bound. Could you please comment on this discrepancy?
> > \
> > \
> > **Additional comments on other points:**
> >
> > **Weakness 1.** The proposed restructuring of the results section sounds good. I would also suggest that the authors improve the explanation of superposition in the initial sections (it was difficult to follow without reading prior work and contributed to some confusion raised in Weakness 2), and add at least a brief discussion on how the proposed explanation of scaling laws connects to existing theories (this is important for placing the work in context and highlighting its novelty, significance, and possible limitations).
> >
> > **Weakness 5.** I appreciate the additional discussion; please consider incorporating it into the appendix. While this could potentially explain why the intuitive explanation and empirical results on mean overlap do not align, it is important for the paper to clearly state that this is the case and acknowledge that the situation is more complex than initially presented.
> >
> > Other points from my review are fully addressed. I would recommend that the authors incorporate the provided clarifications and proposed modifications into the final version of the paper. I also think it would be beneficial to include the small additional experiments discussed in relation to Weakness 2.2 and Additional Concern 10 in the appendix.

---

> > > ### Author Response · Authors · 2025-08-06
> > >
> > > Thank you for the comments. We believe our manuscript will be improved a lot after incorporating the suggested changes.
> > >
> > > **Weakness 3**. Without the assumption, the loss will be something like $\sum_{i \ge m}p_i (1-p_i)^2$. To show that this summation is approximately a power law, or the leading term is a power law, we can use an integral to replace the summation and approximate $1-p_i$ as $1$ since $p_i \ll 1$. Our original writing attempts to use the simplest approach to reach the power law, sacrificing rigor. We can alternatively do rigorous math first, and point out that the loss is approximately a power law at the end, after obtaining the rigorous form of it.
> > >
> > > **Weakness 4**. We think that there are at least three factors making the actual configuration deviate from the ideal ETF. For small $m$ where $m^2/2 < n$, since not all vectors can be put into an ideal ETF, there are two parts of the loss. The first part is from an ETF-like configuration, and the second part comes from the rest of the vectors with larger overlaps. It can be better to put more than $m^2/2$ vectors into the ETF-like configuration to decrease the overall loss, which leads to non-zero but small variance of the overlaps. The second factor is about the loss landscape, which we believe is very rugged. We tried to construct a perfect ETF just by minimizing the overlaps. In the case $m=3$ and $n=6$, where we can visualize the configuration easily, we found that an ETF is hard to reach, and usually we were stuck in some local minimum with non-zero variance but a mean overlap agreeing with $\kappa$ in Eq. (5). It might be true that there are exponentially many local minima with the same mean overlap $\kappa$. The third factor is about the Adam optimizer used, which may not be good at rotating vectors. To gain the big picture, let's use SignGD to approximate Adam. In a two-dimensional optimization example, the true gradient along the two axes can be both positive but different in values, suggesting rotation. Yet, SignGD will make the updates in this case always 45 degrees to each axis, limiting the rotation. We are currently working on the re-parametrization of weight matrices in LLMs and proposing new optimizers.
> > >
> > > There is a separate point we wanted to make, but did not in the main text (to maintain a compact flow). The $\alpha_m =1$ is not just robust to a range of $\alpha$ but also to training. The randomly initialized weight matrix has a mean square overlap of the rows as $1/m$. A surprise is that even after minimizing the overlaps, the best value we can have as $\kappa$ in Eq. (5) is not far away from $1/m$ when $m \ll n$. Our goal is to explain the $1/m$ scaling. And one perspective is that since both random initialization and the delicate ETF yield this same scaling, then the reality (which may be something in between) would naturally have that. In other words, the $1/m$ scaling might be a very generic result (not attached to ETF specifically).
> > >
> > > **Weakness 1**. We totally agree with you. We will try to clearly define the concepts of representation and superposition. Once we have more space in the main text, we can move our Related Works and Discussion sections back, which can highlight the connections to previous theories and our limitations. As a short summary, as far as we can see, most previous theories conceptually fall into our weak superposition regime, whose power laws highly depend on data properties. Our results in the strong superposition regime are complementary to existing ideas, which suggests a mechanism of power laws due to the intrinsic model configuration or limitation.
> > >
> > > **Weakness 5**: Sure! We will definitely address these points.
> > >
> > > We have already done the experiments related to Weakness 2.2 and Concern 10. We will add these results. Thank you.

---

> > > > ### Comment · Reviewer_Y6XA · 2025-08-08
> > > >
> > > > **Weakness 3.** If I am not mistaken, the loss would be $\sum_{i < m} \left(\langle v \rangle_x p_i - p_i^2\right)$ (by the property of variance: $Var(X) = \mathbb{E}(X^2) - \mathbb{E}(X)^2$). I think the current simplification is okay, but I would recommend: 1) discussing the detailed version in the appendix, 2) possibly using the unsimplified loss estimate for Figure 5a. The latter seems important since the actual loss is already a bit higher than the simplified loss estimate, while the non-simplified one would be even lower. But I generally agree that the mean values add a second-order term with respect to $p_i$ here, so the results would most likely not change much.
> > > >
> > > > **Weakness 4.** I appreciate the additional discussion and would recommend adding it to the paper in some way. While the initial claim in the paper that the actual results are very close to the ETF prediction in Figure 6c seemed too bold and not fully supported by the plot, I find the more detailed discussion explaining how it shows similarity to ETF, despite not being exactly ETF, very reasonable.
> > > >
> > > > **Final comment**
> > > >
> > > > The rebuttal and discussion addressed most of my concerns. While some questions remain open, the authors provided reasonable discussion on all of them. I believe that after incorporating these discussions and improving the structure of the text, the paper would make a strong contribution. Therefore, I will increase my score and recommend the paper for acceptance.

---

> > > > > ### Author Response · Authors · 2025-08-08
> > > > >
> > > > > **Weakness 3**. Thank you, you are right. The loss is $\sum_{i\ge m} (\langle v^2\rangle p_i -  \langle v\rangle^2 p_i^2)$, and in our case $\langle v^2\rangle = 4/3$ and $\langle v\rangle = 1$. We will follow the recommendations and compare the simplified and unsimplified losses. There are two reasons why the actual loss is higher than the estimation. The first one is that due to imperfect optimization, not exactly the most important features are represented (Figure 3a, inner bar plot). And the second reason is that lots of data points have superposition (not strong).
> > > > >
> > > > > **Weakness 4**. Thank you, we will add discussions and revise the claim accordingly.
> > > > >
> > > > > We will keep working on the open questions. We appreciate the reviewer's helpful comments and suggestions!

---

### Decision · Program_Chairs · 2025-09-17

**Decision:**

Accept (oral)

**Comment:**

This paper investigates the role of superposition in neural scaling laws. The authors introduce a toy model to examine how superposition and data structure affect the scaling of loss with model size. They find that under strong superposition where all features are represented but overlap, the loss scales robustly as an inverse power law with model dimension. In contrast, weak superposition leads to more fragile scaling behavior.

The reviewers agree that the findings are novel, well-motivated, and supported by careful experimentation. The paper is clearly written, and the analysis illuminates an important yet underexplored mechanism in neural scaling.